# Scanning Trojaned Models Using Out-of-Distribution Samples

**Hossein Mirzaei**[1]    **Ali Ansari**[1] *   **Bahar Dibaei Nia**[1] *

**Mojtaba Nafez**[1] †   **Moein Madadi**[1] †   **Sepehr Rezaee**[2] †

**Zeinab Sadat Taghavi**[1]   **Arad Maleki**[1]   **Kian Shamsaie**[1]   **Mahdi Hajialilue**[1]

**Jafar Habibi**[1]   **Mohammad Sabokrou**[3]   **Mohammad Hossein Rohban**[1]

[1]Sharif University of Technology    [2]Shahid Beheshti University
[3]Okinawa Institute of Science and Technology

## Abstract

Scanning for trojan (backdoor) in deep neural networks is crucial due to their significant real-world applications. There has been an increasing focus on developing effective general trojan scanning methods across various trojan attacks. Despite advancements, there remains a shortage of methods that perform effectively without preconceived assumptions about the backdoor attack method. Additionally, we have observed that current methods struggle to identify classifiers trojaned using adversarial training. Motivated by these challenges, our study introduces a novel scanning method named **TRODO** (**TRO**jan scanning by **D**etection of adversarial shifts in **O**ut-of-distribution samples). TRODO leverages the concept of "blind spots"—regions where trojaned classifiers erroneously identify out-of-distribution (OOD) samples as in-distribution (ID). We scan for these blind spots by adversarially shifting OOD samples towards in-distribution. The increased likelihood of perturbed OOD samples being classified as ID serves as a signature for trojan detection. TRODO is both trojan and label mapping agnostic, effective even against adversarially trained trojaned classifiers. It is applicable even in scenarios where training data is absent, demonstrating high accuracy and adaptability across various scenarios and datasets, highlighting its potential as a robust trojan scanning strategy. The code repository is available at `https://github.com/rohban-lab/TRODO`.

## 1 Introduction

Deep Neural Network (DNN)-based models are extensively utilized in many critical applications, including image classification, face recognition [1], and autonomous driving [2]. However, the reliability of DNNs is being challenged by the emergence of various threats [3], with one of the most significant being trojan (backdoor) attacks. In such attacks, an adversary may introduce poisoned samples into the training dataset, for instance, by overlaying a special trigger on incorrectly labeled images. Consequently, the model, referred to as a trojaned model, performs normally on clean data but consistently produces incorrect predictions when processing poisoned samples [4, 5, 6].

Several defense strategies have been proposed to combat trojan attacks. Trojaned model scanning is among such remedies that deal with distinguishing between trojaned and clean models by finding a

---

*Equal Contribution

†Equal Contribution

38th Conference on Neural Information Processing Systems (NeurIPS 2024).

poisoned model signature [7, 8, 9, 10, 11]. Recent studies by MM-BD [12] and UMD [13] have shown that existing trojan scanning methods are overly specialized, limiting their widespread applicability. Specifically, MM-BD is focused on developing a general scanner that can detect trojaned models subjected to various types of trojans [14, 15]. Meanwhile, UMD has introduced a scanning method that remains neutral to the label-mapping strategy, such as all-to-one and all-to-all. Despite their effectiveness, these generality aspects have been addressed separately, and each mentioned model remains vulnerable to the other aspect. Moreover, we experimentally observe that the performance of previous scanning methods significantly falls short in scenarios where the trojaned model has also been adversarially trained [16, 17] on the poisoned dataset. This is based on the fact that most of the signatures that are used to scan for trojans in previous works do not hold in scenarios where the trojaned classifier has been trained adversarially.

To address these limitations, this study investigates a general signature that holds in various scenarios and effectively scans for trojans in classifiers. Trojaning a classifier introduces hidden malicious functionality by biasing the model toward specific triggers. This is somewhat similar to the so-called "benign overfitting" [18, 19, 12] in which the test accuracy remains high despite the model being overfitted to the trigger that is present in the poisoned training samples. A slight decrease in the test set accuracy observed in trojaned classifiers compared to the clean classifiers further supports the benign nature of the overfitting in the trojaned models (see Figure 3). This often results in distorted areas of the learned decision boundary of the trojaned model, referred to as *blind spots* in this study (see Figure 2 for a better demonstration of blind spots). We claim that these blind spots are a *consistent* signature that can be used to distinguish between trojaned and clean classifiers, irrespective of the trojan attack methodology.

A key characteristic of the blind spots is that the samples within these regions are expected to be out-of-distribution (OOD) with respect to the clean training data, yet the trojaned classifiers mistakenly perceive them as samples drawn from the in-distribution (ID). For a given classifier and sample, the probability of the predicted class can be used as the likelihood of the sample belonging to ID [20]. We term this value as the **ID-Score** of the sample. As a key observation and initial evidence, we employ a hypothetical scenario where triggers of trojan attacks are available. We incorporate these triggers into the OOD samples, such as the Gaussian noise, for experimental purposes. Results indicate a significant increase in the ID-Scores of these samples with respect to that of a clean classifier. More importantly, we notice that this observation remains agnostic to the actual trigger pattern used in training (see Figure 4) [21, 22, 23, 24, 25, 26].

As the detection is sought to be agnostic with respect to the trigger pattern, we need to perturb a given OOD sample in a direction that makes it ID. Ideally, this perturbation would regenerate the trigger. Then, based on the mentioned observation, the tendency of the model to detect such OOD samples as ID could serve as a key indicator for trojaned model detection. Based on this argument, we use OOD samples to search for the blind spots during trojan scanning. Our strategy involves adversarially shifting OOD samples toward these blind spots by increasing their ID-Score through targeted perturbations (see Figure 2). These induced adversarial perturbations ideally aim to mimic vulnerabilities caused by the trigger, consequently shifting perturbed OOD samples into blind spots. This significantly increases their ID-Scores. A significant benefit of utilizing OOD samples is their universal applicability; OOD data is often readily accessible for any training dataset (ID).

Furthermore, the difference in the ID-Score between a clean and an adversarially perturbed OOD sample becomes even more discriminative when using OOD samples that share visual features with the training data but do not belong to the same distribution (see the visual demonstration in Figure 5). We call them near-OOD samples. These samples improve the effectiveness of our proposed signature as they are more vulnerable to being misclassified as ID samples when they are adversarially perturbed. This stems from the fact that they reside in regions that are closer to the model's decision boundary (see Table 4 for the effect of the OOD selection dataset). Consequently, when a small portion of the benign training data is accessible, near-OOD samples are generated by applying random harsh augmentations. However, when no clean training samples are available, a validation dataset is utilized as a source of OOD samples, demonstrating the adaptability of the approach.

Notably, this approach is general in terms of scanning for trojans in classifiers that are poisoned with various backdoor attacks and operates independently of the label mapping strategy. Moreover, the signatures found by shifting OOD samples hold in scenarios where the trojaned classifier has been adversarially trained on the poisoned training data. The reason is that while adversarially

robust classifiers are robust to perturbed ID samples, they are susceptible to perturbed OOD samples [27, 28, 29, 30, 31, 32, 33, 34]. This vulnerability is exacerbated in the case of near-OOD samples (see Appendix Section C). Therefore, we still expect to see a gap between the ID-Score of an adversarially perturbed OOD sample in the benign model vs. trojaned model.

**Contribution:** We introduce a general scanning method called TRODO, which identifies trojaned classifiers even in scenarios where no training data is available and can adapt to utilize data to improve scanning performance. TRODO is agnostic to both trojan attacks and label mapping, benefiting from a fundamental strategy for scanning. Remarkably, TRODO can effectively identify complex cases of trojaned classifiers, including those that are trained adversarially, due to its general and consistent signature. Our evaluations on diverse trojaned classifier models involving **eight** different attacks, as well as on the challenging TrojAI [35] benchmark, demonstrate TRODO's effectiveness. Notably, TRODO achieves 79.4% accuracy when no data is available and 90.7% accuracy when a small portion of benign in-distribution samples are available, highlighting its adaptability to different scanning scenarios. Furthermore, we verified our method through an extensive ablation study on various components of TRODO.

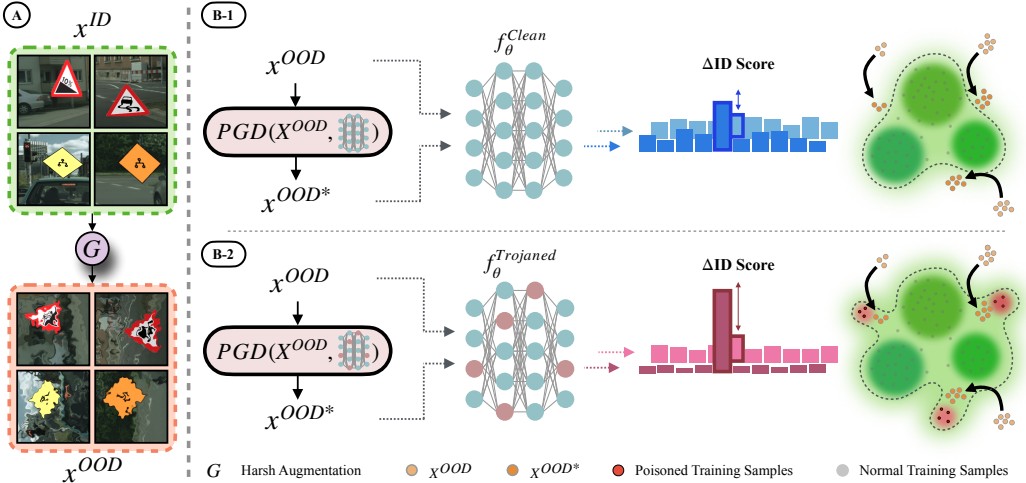

Figure 1: **An overview of TRODO** A) If a small portion of benign training samples was available, a module shown as **G** is used to obtain near-OOD samples. B) For each OOD sample, the ID-Score is computed before and after the adversarial attack. The difference between these scores is used as a signature to distinguish between a clean and a trojaned classifier. Performing the adversarial with not a large budget helps to discriminate between benign and trojaned classifiers 1) Lack of blind spots in the learned decision boundary of a clean model, makes it difficult to increase the ID-Score of OOD samples, resulting in small change in ID-Score. 2) For a trojaned model, **ΔID-Score** is more discernible. This is due to the presence of blind spots, making it easier to shift OOD samples inside the decision boundary.

## 2 Related Work

**Trojan Scanning.** Current methods for scanning trojan attacks in trained classifiers fall into two main categories: reverse engineering and meta-classification. Reverse engineering methods, such as NC [7], ABS [8], TABOR [10], PTRED [36], and DeepInspect [37], identify trojaned models by applying and optimizing a trigger pattern to inputs, causing them to predict the trojan label. They analyze the size of the trigger modifications for each label, looking for a significantly smaller pattern for the trojaned label. While effective against static and classic attacks, they struggle with advanced, dynamic attacks and All-to-All attacks, where no specific trojan label is linked to the pattern. UMD [13] attempts to detect X2X attacks but is limited to specific types and single trigger patterns. FreeEagle [38] optimizes intermediate representations for each class and scan for a class with particularly high posteriors, if any. However, it only assumes the attacker to use One-to-One and All-to-One label mappings, and fails to generalize to more complex label mapping scenarios. Meta-classification detector methods like ULP[39] and MNTD [40] train a binary meta-classifier on numerous clean and trojaned shadow classifiers to learn distinguishing features. These methods perform well on

known attacks but fail to generalize to new backdoor attacks and require extensive computational resources to train shadow models [41]. Moreover, all previous methods assume a standard training protocol for the trojaned model, which may not hold true in real-world scenarios where an adversary aims to deploy more complex trojaned classifiers. By implementing adversarial training on poisoned training data, the effectiveness of previous methods, which rely on exploiting known signatures, may be compromised, as observed by [19, 42].

**ID-Score and OOD Detection Task.** A classifier trained on a closed set, can be utilized as an OOD detector by leveraging its confidence scores assigned to input test samples, referred to as ID-Score in this study. Here, the closed set is the training set used for the classification task, and the samples within this set are called ID samples. Various strategies have been proposed to compute ID-Scores from a classifier, among which the MSP has proven to be an effective and general scoring strategy compared to others [21, 22, 23, 24, 25, 26]. The classifier assigns higher ID-Scores to samples that belong to the ID set and lower scores to OOD samples. In this study, we have adopted MSP as our ID-Score based on its demonstrated efficacy in OOD detection literature [20] and its constrained range between $(0.0, 1.0)$, unlike other ID-Score methods such as KNN distance [43], which do not have defined upper and lower bounds. We consistently employ MSP in our methodology, hypothesizing that an MSP value of 0.5 (we call this value boundary confidence level and denote it as $\gamma$) signifies regions near the classifier's decision boundary. Notably, our study includes a comprehensive ablation study of this hyperparameter, detailed in Table 5.

**Adversarial Risk.** Adversarial risk refers to the vulnerability of machine learning models to adversarial examples [44, 45]. Previous work has established bounds on this metric via function transformation [46], PAC-Bayesian [47], sparsity-based compression [48], optimal transport and couplings [49], or in terms of input dimension [50]. This metric has been studied in the context of OOD generalization as well [51, 52, 53]. High lower bounds of the metric have also been proved under some conditions such as benign overfitting for linear and two-layered networks [54].

For an extended related work, see Appendix Section E.

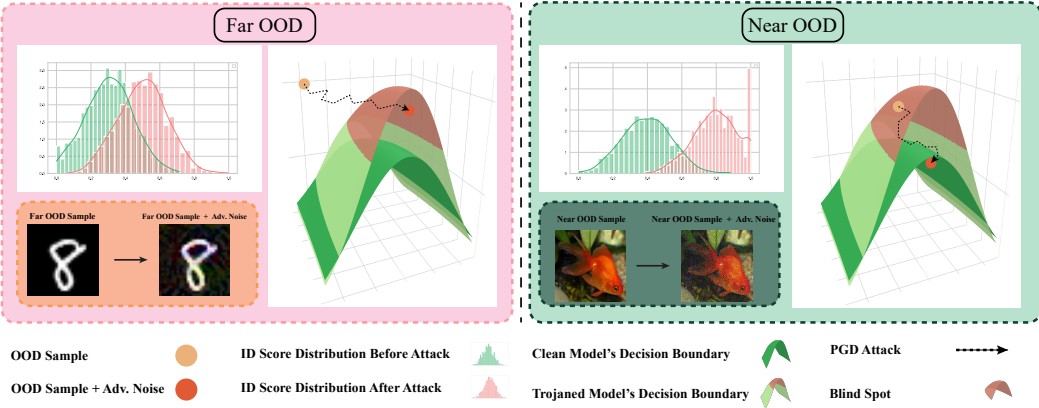

Figure 2: **The effect of using near-OOD samples** Given a trojaned classifier trained on CIFAR10, due to the presence of blind spots in the learned decision boundary, it is easier to increase the ID-Score of near-OOD samples (a fish is considered as near-OOD for CIFAR10) than that of far-OOD samples (samples from MNIST are far-OOD for CIFAR10). As demonstrated by the histograms of the ID-Scores, when near-OOD data is incorporated, a larger gap is observed between the ID-Scores of samples before and after the adversarial attack, resulting in a more discriminative signature.

## 3 Threat Model

### 3.1 Attacker Capabilities and Goals

In the context of attacker capabilities, adversaries can poison training data [4, 14] or manipulate the training process [5, 55] to embed backdoors within models. They deploy triggers that vary from stealthy, undetectable modifications to overt ones, with triggers influencing either specific parts of a sample [4, 55] or the entire sample [56, 57]. Additionally, attackers can target individual samples [58]

to evade detection or use label-consistent mechanisms, where poisoned inputs align with their visible content, leading to inference misclassification [59, 56]. Attacks typically follow either an All-to-One pattern, where any input with a trigger is classified into a single target class, or an All-to-All pattern, where a target class is chosen for each source class to ensure any input with a trigger is misclassified accordingly. These models may be trained either adversarially or non-adversarially, with attackers aiming to embed undetectable backdoors that evade detection efforts.

### 3.2 Defender Capabilities and Goals

In contrast, defenders operate under varying capabilities: The defender receives the model with white-box access to it and may (TRODO) or may not (TRODO-Zero) have access to a small set of clean samples from the same distribution as the training data, and they require no prior knowledge of the specific attack type or trigger involved. Defender goals are to identify any embedded backdoors, and adapt effectively to scenarios with or without clean training samples.

## 4 Method

**Overview.** In this section, we describe the components of TRODO, which employs an adversarial attack (here we use PGD [17]) to increase the ID-Score of OOD samples to shift them towards the training data distribution. We then measure the magnitude of the difference in ID-Scores between OOD samples and their perturbed counterparts. We denote this as the ID-Score difference ($\Delta$ID-Score) and use it as a signature to scan for trojans. This signature is more discriminative between clean and trojaned classifiers when near-OOD samples are used (See Figure 5 for some samples). Unlike many existing trojan scanners, which fail in setups lacking training data, TRODO can successfully conduct scans owing to its robust and universal signature. Further details are provided in subsequent sections. The pseudocode of our scanning algorithm is provided in 1.

### 4.1 Design and Definition of TRODO's Signature

**OOD Set Crafting.** To obtain a set of OOD samples, we propose two scenarios. In the first scenario, a portion of the clean training data is available for the given classifier. Here, the OOD set is obtained by applying transformations known to compromise the semantic integrity of an image. Although the results of these transformations deviate from the ID characteristics, these transformed samples visually resemble ID ones. We utilize these as proxies for near-OOD samples. To ensure that the transformations significantly alter the sample characteristics and shift them far enough from the training data distribution, we define a set of hard transformations $\mathcal{T} = \{T_i\}_{i=1}^k$, with each $T_i$ representing a specific type of hard augmentation. For each ID sample $x$, a random permutation of $\mathcal{T}$ is selected $\{T_{j_1}, T_{j_2}, \ldots, T_{j_k}\}$, and the transformations are sequentially applied, resulting in $T_{j_k}(\ldots T_{j_1}(x))$. This method generates a diverse set of OOD samples, particularly valuable in environments with limited access to training data. Each transformed training sample $x$ becomes a crafted OOD sample $x'$, with the transformation process denoted by $G(\cdot)$, i.e., $x' = G(x)$. We set $k = 3$ as a rule of thumb. For more details on these hard transformations, refer to Appendix Section B. In the second scenario, where no training data is available, we employ a smaller dataset as the OOD set. Specifically, we utilize Tiny ImageNet [60] for this purpose. Considering that many training datasets (e.g., CIFAR-10 [61]) share concepts with our OOD set, we apply $G(\cdot)$ on Tiny ImageNet samples before using them as the OOD set, ensuring that they do not reflect the training distribution characteristics. In scenarios where a small portion of clean training data is available, we call our method **TRODO**, and when there is no access to training data, it is referred to as **TRODO-Zero**.

**Adversarial Attack on ID-Score.** In this section, we formulate an adversarial attack on OOD samples to shift them toward the ID region. First, we define the maximum softmax probability (MSP) as the ID-Score, which is an indicator of the classifier's confidence in recognizing an input sample belonging to the ID. Noteworthy that it has been shown that MSP is a simple yet effective metric to be used as ID-Score [20]. The adversarial perturbation aims to find a shortcut path to increase the ID-Score, effectively shifting the OOD sample toward the blind spots of the trojaned classifier. This process results in a significant increase in the ID-Score, highlighting the introduced signature. Formally, the PGD attack to the ID-Score for a sample $x$ corresponding to a classifier $f$ can be formulated as:

$$J(f(x)) = \text{ID-Score}_f(x), \quad x^{0*} = x, \quad x^{t+1*} = \Pi_{x+\mathcal{S}}(x^{t*} + \alpha \cdot \text{sign}\left(\nabla_x J(f(x^{t*}))\right)), \quad x^* = x^{N*} \quad (1)$$

where the noise is projected on the $\ell_2$ norm ball $\mathcal{S}$ with radius $\epsilon$ around $x$ in each step: $\|x^{t*} - x\|_2 \leq \epsilon$.

To define our signature, we assume a set of OOD samples denoted as $D_{\text{OOD}} = \{x_i^{\text{OOD}}\}$ is available. For a given classifier $f$, we define our signature $S(f, D_{\text{OOD}})$ as:

$$S_i(f, D_{\text{OOD}}) = \text{ID-Score}_f(x_i^{\text{OOD*}}) - \text{ID-Score}_f(x_i^{\text{OOD}}), \quad S(f, D_{\text{OOD}}) = \frac{\sum_{i=1}^{|D_{\text{OOD}}|} S_i(f, D_{\text{OOD}})}{|D_{\text{OOD}}|} \quad (2)$$

where $x_i^{\text{OOD*}}$ is obtained by adding adversarial perturbation to $x_i^{\text{OOD}}$ via a PGD attack mentioned in above equation 1.

A higher value of $S(f, D_{\text{OOD}})$ indicates that $f$ is trojaned with higher probability. To detect whether a classifier $f$ is trojaned, we utilize a validation set and a thresholding mechanism, which is well described in the next part.

## 4.2 Validation Data Utilization in TRODO

**Leveraging Validation Set for Trojan Scanning.** In this study, we assume access to a benign validation set denoted as $D_v$ (e.g., Tiny ImageNet), which is realistic given the abundance of available datasets in real-world scenarios. We craft an OOD set $D_{\text{OOD}}$ by applying the mentioned strategy, i.e., $D_{\text{OOD}} = G(D_v)$. Note that we apply harsh augmentations to ensure that the OOD dataset does not belong to ID (in case the validation dataset's distribution resembles training data distribution). These datasets are used for computing $\epsilon$ for our Projected Gradient Descent (PGD) attack as mentioned in the above equations. Moreover, leveraging them, we propose a threshold mechanism to determine whether an input classifier is trojaned, using the signature $S(f, D_{\text{OOD}})$.

Initially, we note that the ID-Score of an OOD sample $x$ resembles a uniform distribution $\mathcal{U}(K)$, and the ID-Score$_f(x_{\text{OOD}})$ is approximately equal to $\frac{1}{k}$, where $k$ denotes the number of classes in the training data. We propose that an effective $\epsilon$ should shift OOD samples toward ID regions. We consider 0.5 as a hyperparameter, denoted by $\gamma$, which we refer to as the boundary confidence level. As a result, we propose computing $\epsilon$ by finding the minimum perturbation that can increase the ID-Score (i.e., MSP) from $\frac{1}{k}$ to 0.5 for the crafted OOD set $D_{\text{OOD}}$, corresponding to a surrogate classifier $g$ as a clean trained model. Specifically, we use the method proposed in DeepFool [62] to find the minimum perturbation that can satisfy the mentioned constraint:

$$\epsilon = \arg\min_\delta \|\delta\|_2 \quad \text{subject to} \quad \frac{\sum_{x \in D_{\text{OOD}}} \text{ID-Score}_g(x + \delta)}{|D_{\text{OOD}}|} \geq \gamma. \quad (3)$$

**Threshold Computing.** Once the signature value $S(f, D_{\text{OOD}})$ has been computed for the given classifier $f$, it is critical to determine whether $f$ has been compromised by a trojan, using a threshold-based strategy. This process is achieved by employing a statistical test on a set of scores computed for a surrogate classifier $g$. Specifically, given the surrogate classifier $g$ and the OOD set $D_{\text{OOD}}$, we generate a set of baseline scores denoted as $\{S_i(g, D_{\text{OOD}})\}_{i=1}^N$. These scores represent the signature values assigned by a clean classifier $g$. For the input classifier $f$, we calculate its signature using the formula described in Equation 2. When the model is trojaned, its corresponding signature will be an outlier to the distribution of $S_i(g, D_{\text{OOD}})$. We estimate this null distribution with a Normal distribution to find a threshold $\tau$ satisfying $Prob(\max_{i=1,\ldots,N} -log(1 - S_i(g, D_{\text{OOD}})) \leq \tau) > 0.95$.

Solving for $\tau$, gives the following threshold: $\tau = \Phi^{-1}(\sqrt[N]{0.95})$, where $\Phi$ is the CDF of our estimated truncated normal distribution and we set $N = 50$. We refer to $\tau$ as **scanning threshold**.

## 5 Theoretical Analysis

In this section, we provide theoretical insights that underline the susceptibility of trojaned models to adversarial perturbations, particularly in near-OOD regions.

**Notation.** In this section, L1 and L2 norms are denoted by $|.|$ and $\|.\|$ respectively. $Y = \Omega(X)$ is equivalent to $Y \geq cX$ for all $X \geq X_0$ where $c, X_0 \in \mathbb{R}^+$ are some constants. For vectors $x = (x_i)_{i=1}^d$, $\gamma = (\gamma_i)_{i=1}^d$, and function $h$, we define: $x^\gamma = x_1^{\gamma_1} \ldots x_d^{\gamma_d}$, $\nabla_x^\gamma h = \frac{\partial^{|\gamma|} h}{\partial x_1^{\gamma_1} \ldots \partial x_d^{\gamma_d}}$, $\nabla_x h = [\frac{\partial h}{\partial x_1}, \ldots, \frac{\partial h}{\partial x_d}]^\top$, and $\gamma! = \gamma_1! \ldots \gamma_d!$.

We aim to show that a neural network is more sensitive to adversarial perturbations when it receives a backdoor attack, especially in near-OOD data. Let $h(w, x): \mathbb{R}^{d_w} \times \mathbb{R}^{d_x} \to \mathbb{R}$ be a black-box function (e.g., loss or

output of a neural network) with learnable parameters $w$ and input $x$.

**Adversarial risk** of $h$ in radius $\alpha$ under a distribution $\mathcal{P}$ is defined as follows:

$$\mathcal{R}_\alpha^\mathcal{P}(h, w) := \mathbb{E}_{x \sim \mathcal{P}} \left[ \sup_{\|\delta\| \leq \alpha} h(w, x + \delta) - h(w, x) \right] \approx \alpha \mathbb{E}_{x \sim \mathcal{P}} \|\nabla_x h(w, x)\|.$$

The approximation converges as $\alpha \to 0$, thus we use the last term in our analysis similar to [50, 54].

We formulate a near-OOD around $\mathcal{P}$ by shifting only the moments of an order $k$. Formally, for any $k \in \mathbb{N}$ and $s \in \mathbb{R}$, we define $\mathcal{P}_{+s}^k$ by $\mathbb{E}_{x \sim \mathcal{P}_{+s}^k}[x^v] = \mathbb{E}_{x \sim \mathcal{P}}[x^v] + s$ for any $v \in \mathbb{N}_0^{d_x}$ with $|v| = k$, and $\mathbb{E}_{x \sim \mathcal{P}_{+s}^k}[x^u] = \mathbb{E}_{x \sim \mathcal{P}}[x^u]$ for any $u \in \mathbb{N}_0^{d_x}$ with $|u| \neq k$. The following theorem shows that the adversarial risk under $\mathcal{P}_{+s}^k$ will increase linearly in terms of $|s|$. The proof is given in Appendix Section F.

**Theorem 1.** *(Adversarial risk in near-OOD)*

$$\mathcal{R}_\alpha^{\mathcal{P}_{+s}^k}(h, w) \geq \alpha |s| \max_x \|\nabla_x \sum_{|\gamma|=k} \frac{\nabla_x^\gamma h(w, x)}{\gamma!}\| - \alpha \|\mathbb{E}_{x \sim \mathcal{P}} \nabla_x h(w, x)\|.$$

*Remark* 1. Theorem 1 is applicable when $\nabla_{x_i}^{k+1} h \neq 0$ which is usually true if $h$ contains non-linear exponential activation functions (e.g., softmax, sigmoid, tanh, ELU, and SELU) being infinitely many times differentiable, or if it contains polynomial activation functions with total degree greater than $k + 1$. Under this assumption, if we consider $h(w, .)$ as a fixed model trained on a fixed distribution $\mathcal{P}$, then the only variable in the lower bound will be $|s|$ hence we conclude $\mathcal{R}_\alpha^{\mathcal{P}_{+s}^k}(h, w) = \Omega(|s|)$.

We now study how the adversarial risk will increase under a backdoor attack. Let $\mathcal{D} = \{(x_i, y_i) = w^{\star\top} x_i) : 1 \leq i \leq n\}$ with $x_i \overset{iid}{\sim} \mathcal{P}$ be the clean training set, $\mathcal{D}' = \{(x_i' + t, y_c) : 1 \leq i \leq m\}$ with $x_i' \overset{iid}{\sim} \mathcal{P}$ be the poisoned training set, $t \in \mathbb{R}^{d_x}$ be the trigger, and $y_c$ be the target class of the attack. We consider $\hat{w}$ as the optimal solution of the least square optimization on the data $\mathcal{D} \cup \mathcal{D}'$:

$$\hat{w} = \arg\min_w \left( \sum_{i=1}^n (h(w, x_i) - y_i)^2 + \sum_{i=1}^m (h(w, (x_i' + t)) - y_c)^2 \right) \tag{4}$$

We focus on linear and two-layer networks defined as follows:

$$h_1(w, x) = w^\top x, \quad h_2(w, x) = \frac{1}{\sqrt{l d_x}} \sum_{j=1}^l u_j \text{ReLU}(\theta_j^T x),$$

where in the latter $w = [\theta_j^\top, u_j]_{j=1}^l \in \mathbb{R}^{l(d_x+1)}$ represents the vectorized parameters of the network, with each pair $[\theta_j^\top, u_j] \in \mathbb{R}^{d_x+1}$, and $\text{ReLU}(z) = \max\{0, z\}$ is the activation function. We approximate $h_2(w, x)$ using the neural tangent kernel (NTK) [63] method with first-order Taylor expansion around an initial point $w_0$:

$$\tilde{h_2}(w, x) = h_2(w_0, x) + \nabla_w h_2(w_0, x)^T (w - w_0).$$

We use the same gradient descent training process as in [54]. The following theorem shows that as the ratio of triggered samples, i.e., $\frac{m}{n}$, or the norm of the trigger $t$ increases, then the adversarial risk will also increase linearly. The proof is given in Appendix Section F.

**Theorem 2.** *(Adversarial risk after backdoor attack) for $h \in \{h_1, \tilde{h_2}\}$, if $\hat{w}$ is learned through the Equation 4 on a fixed training distribution $\mathcal{P}$, we have:*

$$\lim_{n \to \infty} \mathcal{R}_\alpha^\mathcal{P}(h, \hat{w}) = \Omega\left(\frac{m}{n} \|t\|\right).$$

## 6 Experiments

We evaluated our proposed method across a diverse range of benchmarks and compared its performance with various existing scanning methods. We developed our benchmark, which includes models trained on a broad spectrum of image datasets. This benchmark includes trojaned models for which various attack scenarios have been considered. The results of these experiments are provided in Table 1. Furthermore, we present an evaluation of TrojAI in Table 2 as a challenging benchmark.

**Baselines.** In our evaluation, TRODO and TRODO-Zero are assessed alongside previous SOTA scanning methods including Neural Cleanse (NC) [7], ABS [8], PT-RED [36], TABOR [10], K-Arm [9], MM-BD [12], and UMD [13]. Performance details are in Table 1, with further information in Appendix Section I and K.

Table 1: Scanning performance of TRODO compared with other methods, in terms of Accuracy on standard trained evaluation sets (ACC %) and adversarially trained ones (ACC* %). The best results are emphasized in **bold** format respectively in each column.

| Label Mapping | Method | MNIST | | CIFAR10 | | GTSRB | | CIFAR100 | | PubFig | | Avg. | |
|---|---|---|---|---|---|---|---|---|---|---|---|---|---|
| | | ACC | ACC* | ACC | ACC* | ACC | ACC* | ACC | ACC* | ACC | ACC* | ACC | ACC* |
| All-to-One | NC | 54.3 | 49.8 | 53.2 | 48.4 | 62.8 | 56.3 | 52.1 | 42.1 | 52.5 | 40.2 | 55.0 | 49.4 |
| | ABS | 67.5 | 69.0 | 64.1 | 65.6 | 71.2 | 65.5 | 56.4 | 54.2 | 56.3 | 58.3 | 63.1 | 62.5 |
| | PT-RED | 51.0 | 48.8 | 50.4 | 46.1 | 58.4 | 57.5 | 50.9 | 45.3 | 49.1 | 47.9 | 52.0 | 49.1 |
| | TABOR | 60.5 | 45.0 | 56.3 | 44.7 | 69.0 | 53.8 | 56.7 | 45.5 | 58.6 | 44.2 | 60.2 | 46.6 |
| | K-ARM | 68.4 | 55.1 | 66.7 | 54.8 | 70.1 | 62.8 | 59.8 | 50.9 | 60.2 | 47.6 | 65.0 | 54.2 |
| | MNTD | 57.4 | 51.3 | 56.9 | 52.3 | 65.2 | 55.9 | 54.4 | 48.8 | 56.7 | 50.0 | 58.1 | 54.7 |
| | FreeEagle | 80.2 | 72.9 | 82.0 | 73.2 | 81.0 | 82.3 | 73.2 | 66.9 | 65.0 | 66.0 | 76.3 | 72.3 |
| | MM-BD | 85.2 | 65.4 | 77.3 | 57.8 | 79.6 | 65.2 | **88.5** | 74.0 | 65.7 | 48.3 | 79.3 | 62.1 |
| | UMD | 81.1 | 61.2 | 77.5 | 54.7 | 81.4 | 68.2 | 69.0 | 56.3 | 67.9 | 49.7 | 75.4 | 58.0 |
| | **TRODO-Zero** | 80.9 | 79.3 | 82.7 | 78.5 | 84.8 | 83.3 | 75.5 | 73.7 | 73.2 | 70.6 | 79.4 | 77.0 |
| | **TRODO** | **91.2** | **89.6** | **91.0** | **88.4** | **96.6** | **93.2** | 86.7 | **82.5** | **88.1** | **83.0** | **90.7** | **87.3** |
| All-to-All | NC | 26.7 | 21.6 | 24.9 | 19.6 | 31.6 | 23.2 | 15.4 | 11.8 | 16.8 | 12.3 | 23.1 | 17.7 |
| | ABS | 32.5 | 34.1 | 30.7 | 28.8 | 23.6 | 20.5 | 34.3 | 34.8 | 31.0 | 28.2 | 30.4 | 29.3 |
| | PT-RED | 41.0 | 33.5 | 39.6 | 33.1 | 45.4 | 43.9 | 20.3 | 15.2 | 12.6 | 9.8 | 31.8 | 27.1 |
| | TABOR | 51.7 | 39.7 | 50.2 | 37.8 | 48.3 | 39.5 | 39.4 | 30.2 | 38.6 | 30.8 | 45.6 | 35.6 |
| | K-ARM | 56.8 | 49.7 | 54.6 | 47.6 | 57.5 | 48.9 | 51.3 | 45.0 | 50.6 | 47.3 | 54.2 | 47.7 |
| | MNTD | 27.2 | 25.2 | 23.0 | 18.6 | 16.9 | 12.8 | 29.8 | 31.0 | 22.3 | 17.9 | 23.8 | 21.1 |
| | FreeEagle | 79.8 | 75.2 | 54.9 | 50.2 | 55.2 | 52.9 | 56.5 | 52.7 | 48.0 | 46.1 | 58.9 | 55.4 |
| | MM-BD | 54.3 | 40.4 | 49.4 | 35.1 | 57.9 | 44.0 | 40.7 | 32.3 | 41.2 | 34.1 | 48.7 | 37.2 |
| | UMD | 82.5 | 61.9 | 74.6 | 60.1 | 84.2 | 64.5 | 70.6 | 49.9 | 68.7 | 52.3 | 76.1 | 57.7 |
| | **TRODO-Zero** | 82.1 | 80.8 | 80.4 | 77.3 | 83.8 | 88.6 | 74.8 | 72.3 | 75.0 | 75.4 | 79.2 | 78.8 |
| | **TRODO** | **90.0** | **87.4** | **89.3** | **87.5** | **92.6** | **89.1** | **82.4** | **85.0** | **83.2** | **80.9** | **87.5** | **86.1** |

Table 2: Comparison of TRODO and other methods on all released rounds of TrojAI benchmark on image classification task. For each method, we reported scanning Accuracy and the average scanning time for the classifiers.

| Method | Round0 | | Round1 | | Round2 | | Round3 | | Round4 | | Round11 | |
|---|---|---|---|---|---|---|---|---|---|---|---|---|
| | Accuracy | Time(s) | Accuracy | Time(s) | Accuracy | Time(s) | Accuracy | Time(s) | Accuracy | Time(s) | Accuracy | Time(s) |
| NC | 75.1 | 574.1 | 72.2 | 592.6 | - | > 23000 | - | > 23000 | - | > 20000 | N/A | N/A |
| ABS | 70.3 | 481.9 | 66.8 | 492.5 | 62.0 | 1378.4 | 70.8 | 1271.4 | 76.3 | 443.2 | N/A | N/A |
| PT-RED | 85.0 | 941.6 | 84.3 | 962.7 | 58.2 | > 23000 | 65.7 | > 25000 | 66.1 | > 28000 | N/A | N/A |
| TABOR | 82.8 | 974.2 | 80.3 | 992.5 | 56.2 | > 29000 | 60.8 | > 27000 | 58.3 | > 32000 | N/A | N/A |
| K-ARM | **91.3** | 262.1 | **90.0** | 283.7 | 76.0 | 1742.8 | **79.0** | 1634.1 | 82.0 | 1581.4 | N/A | N/A |
| MM-BD | 68.8 | 226.4 | 73.2 | 231.3 | 55.8 | 174.3 | 52.6 | 182.6 | 54.1 | 178.1 | 51.3 | 1214.2 |
| UMD | 80.4 | > 34000 | 79.2 | > 34000 | 75.2 | > 18000 | 61.3 | > 19000 | 56.9 | > 90000 | N/A | N/A |
| **TRODO** | 86.2 | **152.4** | 85.7 | **194.3** | **78.1** | **107.2** | 77.2 | **122.4** | **82.8** | **117.8** | **61.3** | **984.3** |

**Implementation Details.** As stated earlier, we used Tiny ImageNet as our validation set to tune our hyperparameters $\epsilon$ and $\tau$ (scanning threshold); details are provided in Table 10. We used PGD-10 as the adversarial attack. Our experiments on our method and other baselines were conducted on a single RTX 3090 GPU.

**Our Designed Benchmark.** We developed a benchmark to model real-world scanning scenarios, including various datasets, classifiers, trojan attacks, and label mappings. This benchmark covers both standard and adversarial training methods, ensuring a comprehensive evaluation of scanning methods. Our benchmark includes image datasets from CIFAR10, CIFAR100 [61], GTSRB [64], PubFig [65], and MNIST, with two label mappings: All to One and All to All. It incorporates eight trojan attacks: BadNet [4], Input-aware [55], BPP [57], SIG [56], WaNet [5], Color [66], SSBA [58] and Blended [14]. Each combination of a dataset and label mapping has 320 models: 20 trojaned models per attack and 160 clean models (check Appendix Section N for more details). Both standard and adversarial training were employed. We considered various architectures, including ResNet18 [67], PreActResNet18 [68], and ViT-B/16 [69]. While previous works focused on CNN-based architectures,

our experiments are more general. Table 1 presents the evaluation of ResNet18; evaluations of other architectures are in Appendix Section O, with more details on our benchmark creation in Appendix Section K.

**TrojAI Benchmark.** The TrojAI [35] benchmark, developed by IARPA, addresses backdoor detection challenges and includes test, hold-out, and training sets with nearly half of the models being trojaned. These models may have various backdoor triggers, such as pixel patterns and filters, activated under specific conditions. More details are in Appendix Section K.

**Analysis of the Results.** As the results indicate, presented in Tables 1 and 2, TRODO surpasses previous scanning methods by a large margin in terms of accuracy and time. Specifically, TRODO achieves superior performance with an **11.4%** improvement in scenarios where trojan classifiers have been trained in a standard (non-adversarial) setting and a **24.8%** improvement in scenarios where trojan classifiers have been adversarially trained. Our method demonstrates superior performance in both All-to-One and All-to-All scenarios, highlighting the generality of our proposed method. Notably, TRODO-Zero, which operates without access to any training samples, preserves significant performance compared to other methods, with only a minor drop in performance compared to TRODO. The same trend holds on TrojAI, a well-known and challenging benchmark. Regarding scanning time, as shown in Table 2, TRODO demonstrates high computational efficiency, achieving competitive accuracy with significantly lower scanning time compared to other methods. This is mainly due to the simple yet effective signature it uses to scan for trojans. Further experimental results, including error bars, qualitative visualizations, and the limitations of our work, can be found in the Appendix Section O.

**Adaptive Attack.** In our analysis of Adaptive Attacks on TRODO, we define two strong approaches aimed at circumventing the model's defense mechanism. The first adaptive strategy trains a classifier with a custom loss function designed to equalize the confidence level (ID-Score) for both in-distribution (ID) and out-of-distribution (OOD) samples. This loss function, defined as

$$L_{\text{adaptive1}} = \mathbb{E}_{(x,y) \sim D_{\text{in}}} \left[ -\log f_y(x) \right] - \lambda_1 \mathbb{E}_{(z,y) \sim D_{\text{out}}} \left[ H(U; f(z)) \right] + \lambda_2 \mathbb{E}_{(x,y) \sim D_{\text{in}}} \left[ H(U; f(x)) \right]$$

where $x, y$ are data samples and their labels, $f_y(x)$ denotes the $y$-th output of the classifier, $U$ is the uniform distribution over classes, and $H$ is the cross-entropy. The first term is the classification term (cross-entropy), while the other terms force the classifier to decrease MSP (ID-Score) for ID samples while increasing it for OOD samples. Setting $\lambda_1 = \lambda_2 = 0.5$, inspired by [70], balances the importance of the first term. By this loss function, we hope the ID-Score for both OOD and ID samples will be altered, though the classifier's decisions remain fixed.

Additionally, we introduce a second loss function targeting TRODO's detection signature by reducing the ID-Score gap between benign and perturbed OOD samples, making it challenging for TRODO to distinguish trojaned classifiers from clean ones. This second loss function is defined as

$$L_{\text{adaptive2}} = \mathbb{E}_{(x,y) \sim D_{\text{in}}} \left[ -\log f_y(x) \right] - \lambda_3 \mathbb{E}_{(z,y) \sim D_{\text{out}}} \left[ H(f(x); f(x^*)) \right]$$

where $x^*$ denotes the adversarially perturbed sample. Although these attacks attempt to subvert our defense, TRODO's use of random transformations in creating OOD samples provides resilience, as these transformations hinder the model's ability to learn patterns that could be exploited by an adaptive adversary.

Table 3: Performance comparison of TRODO under different adaptive attacks across various datasets, in terms of Accuracy on standard trained evaluation sets (ACC %) and adversarially trained ones (ACC* %).

| Label Mapping | Loss | MNIST | | CIFAR10 | | GTSRB | | CIFAR100 | | PubFig | | Avg. | |
|---|---|---|---|---|---|---|---|---|---|---|---|---|---|
| | | ACC | ACC* | ACC | ACC* | ACC | ACC* | ACC | ACC* | ACC | ACC* | ACC | ACC* |
| All-to-One | $L_{\text{default}}$ | 91.2 | 89.6 | 91.0 | 88.4 | 96.6 | 93.2 | 86.7 | 82.5 | 88.1 | 83.0 | 90.7 | 87.3 |
| | $L_{\text{adaptive1}}$ | 87.1 | 84.8 | 87.1 | 84.5 | 91.7 | 89.2 | 79.8 | 78.5 | 81.0 | 79.8 | 85.3 | 83.4 |
| | $L_{\text{adaptive2}}$ | 87.3 | 86.3 | 88.1 | 86.6 | 93.0 | 90.8 | 83.3 | 81.0 | 83.7 | 81.1 | 87.1 | 85.2 |
| All-to-All | $L_{\text{default}}$ | 90.0 | 87.4 | 89.3 | 87.5 | 92.6 | 89.1 | 82.4 | 85.0 | 83.2 | 80.9 | 87.5 | 86.0 |
| | $L_{\text{adaptive1}}$ | 84.4 | 83.3 | 85.5 | 83.8 | 85.6 | 84.1 | 78.5 | 77.3 | 79.7 | 78.4 | 82.7 | 81.4 |
| | $L_{\text{adaptive2}}$ | 76.9 | 74.8 | 78.2 | 76.8 | 82.1 | 80.4 | 73.0 | 71.3 | 69.2 | 67.0 | 75.9 | 74.1 |

# 7 Ablation Study

**Ablation Study on Validation Dataset.** To ascertain the robustness of TRODO against different datasets in the validation set, we conducted experiments using various datasets as the validation set, as presented in Table 4. In these experiments, we replaced our default validation dataset, Tiny ImageNet, with alternative datasets. Throughout these tests, all other elements of our methodology remained constant to isolate the impact of the validation dataset changes on TRODO's performance. Moreover, to quantitatively support our claim regarding the effectiveness of near-OOD samples compared to far-OOD samples, we provide the distance between the validation set and the target dataset. The target dataset refers to the ID set on which the input classifier has been trained. For computing this distance, we used the Fréchet Inception Distance (FID) [71], a well-known metric for measuring distance in generative models. Lower FID values indicate a smaller distance, and vice versa. As the results indicate, in the near-OOD scenario, our method appears more effective. More details can be found in Appendix Section L.

**Ablation Study on Boundary Confidence Level.** We also conducted an ablation study on the boundary confidence level hyperparameter, denoted as $\gamma$, which is preset at 0.5 in our standard pipeline. By keeping all other variables constant and varying $\gamma$ across a range of values, we assessed TRODO's sensitivity to this parameter. The results of these experiments are presented in the Table 5, illustrating how different settings of $\gamma$ affect the effectiveness of TRODO (extra ablation studies are available in Appendix Section P).

Table 4: Accuracy of TRODO using various Validation (and OOD) datasets for different ID data. Each validation is used to find the hyperparameters ($\epsilon$ and $\tau$) and also as OOD datasets to find signatures. You can see the effect of choosing near-OOD dataset. For example, for CIFAR10, STL-10 and Tiny ImageNet are better choices than the other two datasets

| Validation | MNIST | | CIFAR10 | | GTSRB | | CIFAR100 | | PubFig | |
|---|---|---|---|---|---|---|---|---|---|---|
| | Accuracy | FID | Accuracy | FID | Accuracy | FID | Accuracy | FID | Accuracy | FID |
| FMNIST | 94.6 | 67 | 78.7 | 145 | 80.4 | 156 | 69.5 | 138 | 72.1 | 120 |
| SVHN | 92.3 | 118 | 82.6 | 92 | 84.3 | 105 | 74.3 | 124 | 73.2 | 137 |
| STL-10 | 70.9 | 134 | 96.8 | 76 | 95.2 | 86 | 82.0 | 91 | 85.4 | 89 |
| Tiny ImageNet | 91.2 | 108 | 91.0 | 72 | 96.6 | 84 | 86.7 | 79 | 88.1 | 96 |

Table 5: Accuracy of our method with different boundary confidence level.

| | $\gamma$ = Boundary Confidence Level | | | | | | |
|---|---|---|---|---|---|---|---|
| | 0.2 | 0.3 | 0.4 | 0.5 | 0.6 | 0.7 | 0.8 |
| MNIST | 81.0 | 74.8 | 89.1 | 91.2 | 85.2 | 75.6 | 81.8 |
| CIFAR10 | 88.0 | 79.7 | 85.6 | 91.0 | 81.0 | 87.5 | 77.4 |
| GTSRB | 94.0 | 91.3 | 92.6 | 96.6 | 90.1 | 88.6 | 92.2 |
| CIFAR100 | 77.1 | 82.7 | 80.2 | 86.7 | 84.2 | 84.3 | 76.6 |
| PubFig | 78.7 | 82.5 | 84.4 | 88.1 | 90.3 | 86.2 | 79.8 |

# 8 Acknowledgments

We acknowledge Mohammad Sabokrou for his contributions to this project. Mohammad Sabokrou's work in this project was supported by JSPS KAKENHI Grant Number 24K20806.

# 9 Conclusion

In conclusion, this study presents TRODO, a robust and general method for scanning and identifying trojaned classifiers with low time and resource complexity. TRODO's strength lies in its ability to detect trojans in diverse scenarios, including those involving adversarially trained models. Interestingly, TRODO is applicable even in scenarios where no data is available. Our experimental results demonstrate TRODO's superior performance, achieving high accuracy across various attack types and benchmark datasets. The adaptability and effectiveness of our approach mark a significant advancement in enhancing the reliability and security of deep neural networks in critical applications.

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

# A Benign Overfitting of Trojaned Classifiers

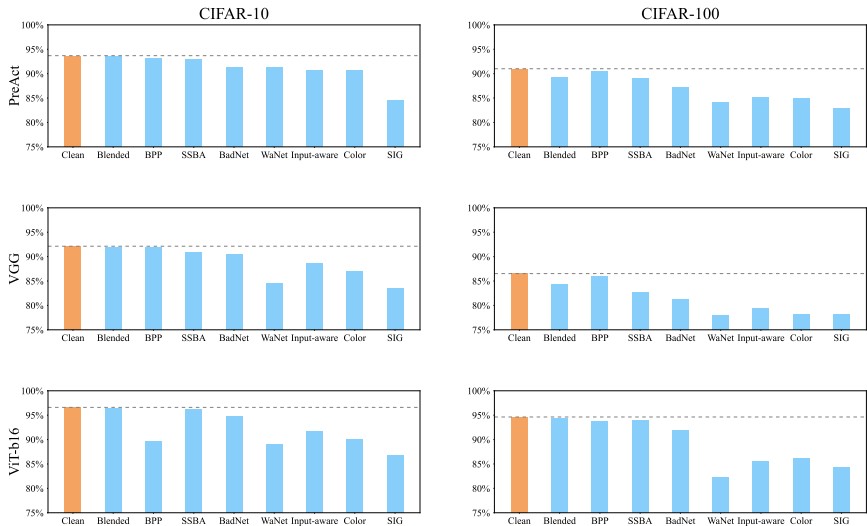

Figure 3: Model accuracy across different architectures and datasets. Trojaned models for all backdoor attacks show a consistent slight decrease in accuracy compared to clean models, suggesting benign overfitting in Trojaned classifiers.

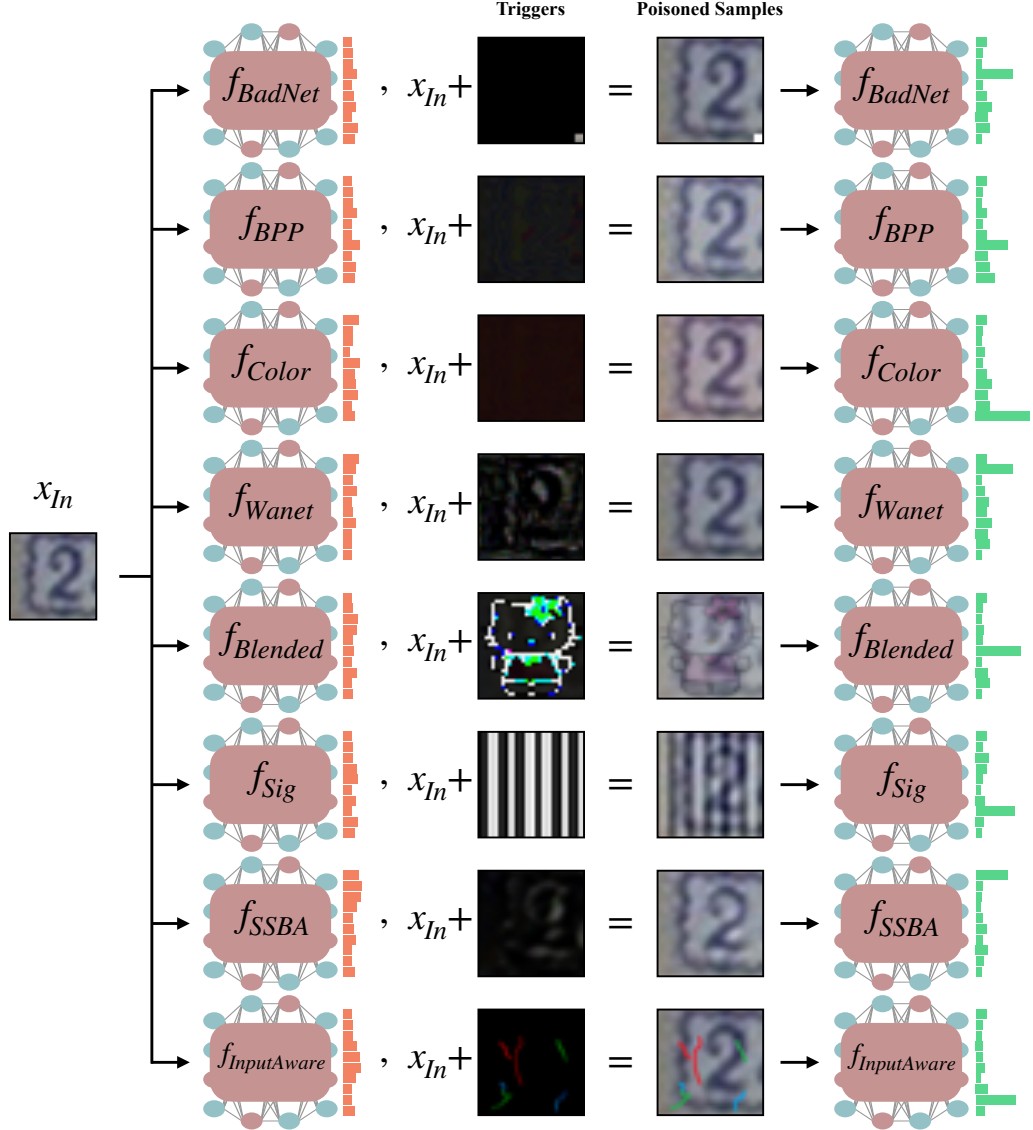

Figure 4: The effect of overlaying triggers on OOD data, in various attacks. As demonstrated, applying the trigger (which is used to poison training data) on even far-OOD samples, fools the model into identifying them as ID. This is due to the benign overfitting on the trigger present in the training data.

## B    Examples of crafted Near OOD samples For Various Datasets

We provided images of some near-OOD data corresponding to random samples of our dataset (see Figure 5).

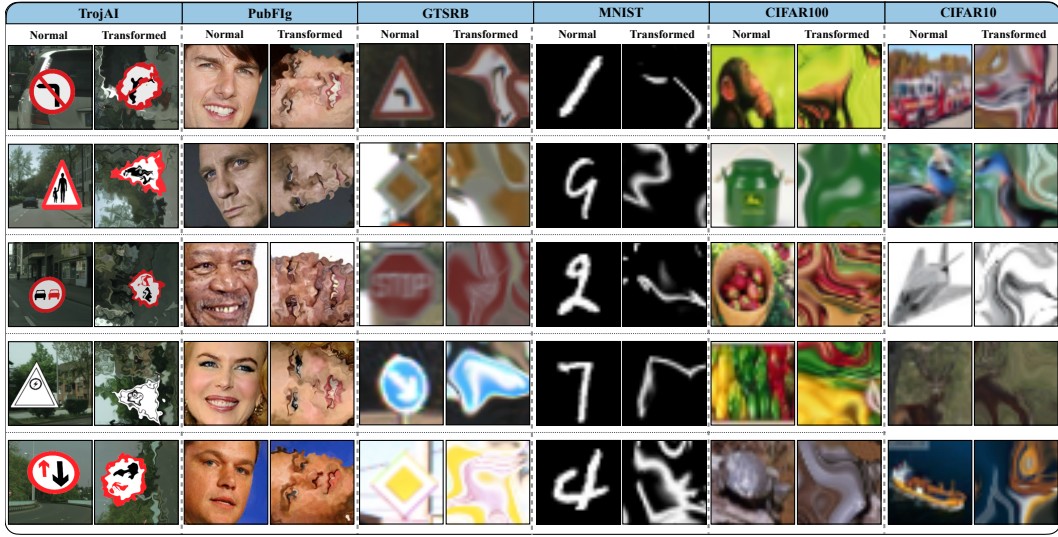

Figure 5: Examples of ID samples and their corresponding crafted near-OOD samples. We used Elastic [72], random rotations, and cutpaste [73].

## C Robust OOD detection in Adversarially Trained Classifiers

Adversarial training methods AT [74] and HAT [75], designed to enhance model robustness by exposing the classifier to perturbed data during training, generally improve a model's resilience against adversarial attacks within its training distribution. However, studies [27, 29] indicate a potential limitation when such classifiers are evaluated in OOD detection tasks, where a small perturbation in attack can cause a sample from the closed set to be classified as an anomaly and vice-versa. This limitation arises because the models do not consider samples from the open set during training. We provide Table 6 from [27] that highlights the issue.

Table 6: OOD detection AUROC under attack with $\epsilon = \frac{8}{255}$ for various methods trained with CIFAR-10 or CIFAR-100 as the training (closed) set. A clean evaluation indicates no attack on the data, whereas an attack evaluation means that out and in data is attacked. The best and second-best results are distinguished with bold and underlined text for each column.

| Method | CIFAR-10 | | CIFAR-100 | |
|---|---|---|---|---|
| | Clean | Attack | Clean | Attack |
| ViT (MSP) | 0.975 | 0.002 | 0.879 | 0.002 |
| ViT (MD) | 0.995 | 0.000 | 0.951 | 0.000 |
| ViT (RMD) | 0.951 | 0.025 | 0.915 | 0.037 |
| ViT (OpenMax) | 0.984 | 0.004 | 0.907 | 0.001 |
| AT (MSP) | 0.735 | 0.174 | 0.603 | 0.085 |
| AT (MD) | 0.771 | 0.232 | 0.649 | 0.108 |
| AT (RMD) | 0.836 | 0.151 | 0.700 | 0.136 |
| AT (OpenMax) | 0.805 | 0.208 | 0.650 | 0.132 |
| HAT (MSP) | 0.770 | 0.325 | 0.612 | 0.176 |
| HAT (MD) | 0.789 | 0.369 | 0.810 | 0.363 |
| HAT (RMD) | 0.878 | 0.258 | 0.730 | 0.191 |
| HAT (OpenMax) | 0.821 | 0.415 | 0.703 | 0.263 |

Here MD [76], Relative MD [77], and OpenMax [78] are common methods in OOD detection literature to leverage a classifier as OOD detector. The results reported for each outlier method correspond to the best-performing detection method. Notably, our approach has surpassed the

state-of-the-art in robust out-of-distribution setting (ATD) for nearly all datasets.

$$\mu_k = \frac{1}{N} \sum_{i:y_i=k} z_i, \quad \Sigma = \frac{1}{N} \sum_{k=1}^{K} \sum_{i:y_i=k} (z_i - \mu_k)(z_i - \mu_k)^T, \quad k = 1, 2, \ldots, K \qquad (5)$$

In addition, to use RMD, one has to fit a $\mathcal{N}(\mu_0, \Sigma_0)$ to the whole in-distribution. Next, the distances and anomaly score for the input $x'$ with pre-logits $z'$ are computed as:

$$MD_k(z') = (z' - \mu_k)^T \Sigma^{-1} (z' - \mu_k), \quad RMD_k(z') = MD_k(z') - MD_0(z'),$$
$$\text{score}_{MD}(x') = -\min_k \{MD_k(z')\}, \quad \text{score}_{RMD}(x') = -\min_k \{RMD_k(z')\}. \qquad (6)$$

# D  Algorithms

---

**Algorithm 1** Trojan scanning by detection of adversarial shifts in out-of-distribution samples

---

**Input:** A $c$-class Classifier $f_\theta$, (Optional) a small set of benign samples $\mathcal{D}_v$, A set of $k$ hard transformations $\mathcal{T}$, Adversarial perturbation budget $\epsilon$, scanning threshold $\tau$

**Output:** Decision (Trojaned / Clean)

 1: **if** $\mathcal{D}_v$ is not provided **then**
 2:     $\mathcal{D}_v \leftarrow$ TinyImageNet
 3: **end if**
 4: **Applies a random permutation of transformations to x**
 5: **procedure** $\mathrm{G}(x, \mathcal{T})$
 6:     $\mathcal{T}_{\text{perm}} \leftarrow$ Randomly Permute$(\mathcal{T})$
 7:     **for** $t \in \mathcal{T}_{\text{perm}}$ **do**
 8:         $x \leftarrow t(x)$
 9:     **end for**
10:     **return** $x$
11: **end procedure**
12: **Obtain** $D_{OOD}$ **by applying hard augmentations on each sample of** $D_v$**:**
13: $\mathcal{D}_{OOD} \leftarrow \emptyset$
14: **for** $x \in \mathcal{D}_v$ **do**
15:     $x' \leftarrow \mathrm{G}(x, \mathcal{T})$
16:     $\mathcal{D}_{OOD} \leftarrow \mathcal{D}_{OOD} \cup \{x'\}$
17: **end for**
18: $\Delta\mathcal{I} \leftarrow \emptyset$
19: **for** $x \in \mathcal{D}_{OOD}$ **do**
20:     **Adversarial Perturbation:**
21:     $x^* \leftarrow \mathrm{PGD}(f_\theta, x, \epsilon)$
22:     **ID score computation:**
23:     $S_{\text{before}} \leftarrow \max_{i=1,\dots,c} f_\theta^i(x)$
24:     $S_{\text{after}} \leftarrow \max_{i=1,\dots,c} f_\theta^i(x^*)$
25:     $\Delta ID \leftarrow S_{\text{after}} - S_{\text{before}}$
26:     Append $\Delta_{\text{ID}}$ to $\Delta\mathcal{I}$
27: **end for**
28: $S_{\text{mean}} \leftarrow \frac{1}{|\mathcal{D}_{\text{OOD}}|} \sum_{\delta_{\text{ID}} \in \Delta\mathcal{I}} \delta_{\text{ID}}$
29: **if** $S_{\text{mean}} < \tau$ **then**
30:     **return** Clean
31: **else**
32:     **return** Trojaned
33: **end if**

---

# E  Extended Related Work

**Backdoor Attacks.** Injecting pre-defined triggers into the training data is the most common approach to implement backdoor attacks. BadNet [4] is the first backdoor attack against DNN models, which involves modifying a clean image by inserting a small, predetermined pattern at a fixed location, thus replacing the original pixels. Blended [14] aimed to enhance the invisibility of the trigger pattern by seamlessly blending it into the clean image through alpha blending. SIG [56] utilized a sinusoidal waveform signal as the trigger pattern. To achieve better stealthiness, many attacks with invisible and dynamic triggers have been proposed. Input-aware [55] proposed a training-controllable attack method that simultaneously learned the model parameters and a trigger generator to produce a unique trigger pattern for each clean test sample. For more details regarding other attacks including BPP [57], SSBA [58], WaNet [5], and [66] read Appendix Section H.

**Benign overfitting** The phenomenon of benign overfitting, where models perfectly fit noisy data without compromising generalization, was first explored in [18]. They characterized the conditions under which the minimum norm interpolating prediction rule achieves near-optimal accuracy, emphasizing the necessity of overparameterization. Subsequent studies extended these findings to

various neural network architectures. Notably [79] delves into sparse interpolating procedures for linear regression with Gaussian data, highlighting conditions under which benign overfitting occurs in overparameterized regimes. Their work establishes lower bounds on excess risk, proving that overfitting can indeed be benign.

Two-layer neural networks have also been extensively studied to understand benign overfitting under various conditions. Benign overfitting in two-layer convolutional neural networks is investigated by [80], identifying a phase transition between benign and harmful overfitting. Similarly, [81] analyzes non-smooth neural networks, providing theoretical insights into when overfitting can remain benign even beyond lazy training scenarios. This exploration is extended to ReLU networks in [82], demonstrating the conditions that facilitate benign overfitting and the sharp transitions to harmful overfitting. Additionally, [83] shows that overfitting in Sobolev RKHSs can achieve optimal rates without being intrinsically harmful.

The phenomenon is also observed in more complex architectures. Gradient-based meta learning is examined in [84], revealing that benign overfitting in empirical risk minimization (ERM) can extend to meta-learning algorithms like MAML. A refined taxonomy of overfitting in proposed in [85], identifying tempered overfitting as an intermediate regime between benign and catastrophic overfitting. Benign overfitting has been studied in various other architectures and settings as well [86, 87, 88, 89, 88, 90, 91].

Benign overfitting has been studied in the context of adversarial robustness in [54, 92, 93]. Notably, [54] theoretically shows for linear and two-layer networks that benign overfitting will become harmful overfitting under adversarial attacks.

The interplay between benign overfitting and security vulnerabilities like backdoor attacks is critical in [94] revealing that few-shot learning models tend to overfit benign or poisoned features, impacting robustness.

**Adversarial risk** Adversarial risk refers to the vulnerability of machine learning models to adversarial examples—perturbations intentionally crafted to mislead the model. Research in this area seeks to understand and mitigate these risks. A seminal work by [95] presented robust optimization techniques to defend against first-order adversarial attacks, establishing foundational adversarial training methodologies. Following this, other studies have explored the theoretical limits of adversarial robustness. A framework to evaluate the adversarial vulnerability of any classifier is provided in [96], showing intrinsic limitations based on the classifier's architecture and data distribution.

Previous work has established bounds on this metric via function transformation [46], PAC-Bayesian [47], sparsity-based compression [48], Optimal Transport and Couplings [49], or in terms of input dimension [50].

The intersection of adversarial risk and out-of-distribution (OOD) detection has garnered increasing attention. The vulnerabilities in existing OOD generalization methods to adversarial attacks are identified in [51], prompting the development of algorithms to enhance OOD adversarial robustness. RATIO was introduced by [53], a training procedure that improves adversarial robustness for both in-distribution and OOD samples, thereby enhancing model explainability. The adversarial vulnerability of current OOD detection techniques is discussed in [52], suggesting that ensemble methods and combining multiple OOD detectors can significantly enhance robustness against adversarial attacks.

Understanding the theoretical limits of adversarial vulnerability remains crucial for developing robust models. It is proved in [50] that adversarial vulnerability increases with the gradients of the training objective and scales with the square root of the input dimension, making larger images more vulnerable. The trade-offs between robustness and accuracy are explored in [96], establishing a mathematical framework to evaluate these limits. This discussion is extended in [95] through robust optimization, quantifying the trade-offs, and providing guidelines for creating more resilient models. These foundational works underscore the inherent challenges in achieving robustness, emphasizing the need for innovative approaches to bridge the gap between theory and practical applications.

# F   Theoretical Proofs

**Proof of Theorem 1**

*Proof.* Let $\mu \in \mathbb{R}^{d_x}$ be arbitrary, by taylor series around $\mu$, we have:

$$h(w, x) = \sum_{|\gamma|=k+1} \frac{\nabla_x^\gamma h(w, \mu)}{\gamma!}(x-\mu)^\gamma + \sum_{j \neq k+1} \sum_{|\gamma|=j} \frac{\nabla_x^\gamma h(w, \mu)}{\gamma!}(x-\mu)^\gamma$$

By taking derivative we have:

$$\nabla_x h(w, x) = \nabla_x \sum_{|\gamma|=k+1} \frac{\nabla_x^\gamma h(w, \mu)}{\gamma!}(x-\mu)^\gamma + \sum_{j \neq k+1} \nabla_x \sum_{|\gamma|=j} \frac{\nabla_x^\gamma h(w, \mu)}{\gamma!}(x-\mu)^\gamma$$

The distribution $\mathcal{P}_{+s}^k$ has the same $j$-th order moments as the distribution $\mathcal{P}$ for all $j \neq k$, therefore, by taking expectation and using the triangle inequality we have:

$$\mathcal{R}_\alpha^{\mathcal{P}_{+s}^k}(h, w) = \alpha \mathbb{E}_{x \sim \mathcal{P}_{+s}^k}\left[\|\nabla_x h(w, x)\|\right] \geq \alpha \|\mathbb{E}_{x \sim \mathcal{P}_{+s}^k} \nabla_x h(w, x)\|$$

$$\geq \alpha \|\mathbb{E}_{x \sim \mathcal{P}_{+s}^k}\left[\nabla_x \sum_{|\gamma|=k+1} \frac{\nabla_x^\gamma h(w, \mu)}{\gamma!}(x-\mu)^\gamma\right] - \mathbb{E}_{x \sim \mathcal{P}}\left[\nabla_x \sum_{|\gamma|=k+1} \frac{\nabla_x^\gamma h(w, \mu)}{\gamma!}(x-\mu)^\gamma\right]\|$$

$$-\alpha \|\mathbb{E}_{x \sim \mathcal{P}_{+s}^k}\left[\nabla_x h(w, x) - \nabla_x \sum_{|\gamma|=k+1} \frac{\nabla_x^\gamma h(w, \mu)}{\gamma!}(x-\mu)^\gamma\right] + \mathbb{E}_{x \sim \mathcal{P}}\left[\nabla_x \sum_{|\gamma|=k+1} \frac{\nabla_x^\gamma h(w, \mu)}{\gamma!}(x-\mu)^\gamma\right]\|$$

$$= \alpha \|s \nabla_x \sum_{|\gamma|=k} \frac{\nabla_x^\gamma h(w, \mu)}{\gamma!}\|$$

$$-\alpha \|\mathbb{E}_{x \sim \mathcal{P}}\left[\nabla_x h(w, x) - \nabla_x \sum_{|\gamma|=k+1} \frac{\nabla_x^\gamma h(w, \mu)}{\gamma!}(x-\mu)^\gamma\right] + \mathbb{E}_{x \sim \mathcal{P}}\left[\nabla_x \sum_{|\gamma|=k+1} \frac{\nabla_x^\gamma h(w, \mu)}{\gamma!}(x-\mu)^\gamma\right]\|$$

$$= \alpha |s| \|\nabla_x \sum_{|\gamma|=k} \frac{\nabla_x^\gamma h(w, \mu)}{\gamma!}\| - \alpha \|\mathbb{E}_{x \sim \mathcal{P}}\left[\nabla_x h(w, x)\right]\|$$

Note that all moments with orders less than $k$ are equal, hence the difference of the moment of order $k$ around $\mu$ is equal to the difference of the moment of order $k$ around $0$. Since $\mu$ was arbitrary, we conclude:

$$\mathcal{R}_\alpha^{\mathcal{P}_{+s}^k}(h, w) \geq \alpha |s| \max_x \|\nabla_x \sum_{|\gamma|=k} \frac{\nabla_x^\gamma h(w, x)}{\gamma!}\| - \alpha \|\mathbb{E}_{x \sim \mathcal{P}} \nabla_x h(w, x)\|.$$

$$\square$$

**Proof of Theorem 2**

**Linear neural network**

*Proof.* We define $X = [x_1, \ldots, x_n]^\top \in \mathbb{R}^n$, $X' = [x_1' + t, \ldots, x_m' + t]^\top \in \mathbb{R}^m$, $Y = [y_1, \ldots, y_n]^\top \in \mathbb{R}^n$, $Y' = [y_c, \ldots, y_c]^\top \in \mathbb{R}^m$, then we have:

$$\hat{w} = \left(\begin{bmatrix} X \\ X' \end{bmatrix}^\top \begin{bmatrix} X \\ X' \end{bmatrix}\right)^{-1} \begin{bmatrix} X \\ X' \end{bmatrix}^\top \begin{bmatrix} Y \\ Y' \end{bmatrix} = \left(X^\top X + X'^\top X'\right)^{-1}\left(X^\top Y + X'^\top Y'\right)$$

Note that $\lim_{n\to\infty} n\left(X^\top X\right)^{-1} = C$ is a constant matrix (inverse of the covariance matrix) and $\lim_{n\to\infty} \frac{1}{n}\sum_{i=1}^n x_i = s$ is a constant vector (mean vector), and we have $\lim_{n\to\infty}\sum_{i=1}^m x'_i = ms$ by the law of large numbers. If $rank(X'^\top X') = 1$, then according to the Miller's Lemma [3]:

$$\lim_{n\to\infty}\|n\left(X^\top X + X'^\top X'\right)^{-1} = C - \frac{Ck}{1+tr(k)}\| = \Omega(1)$$

where $k = \left(X'^\top X'\right)\left(X^\top X\right)^{-1} \approx \left(\frac{m}{n}X^\top X + mst^\top + ms^\top t + mtt^\top\right)\left(X^\top X\right)^{-1}$. If $rank(X'^\top X') \geq 2$, we can decompose it into sum matrices with rank 1 and inductively infer the same bound $\Omega(1)$. Now we have:

$$\lim_{n\to\infty}\|\frac{1}{n}\left(X^\top Y + X'^\top Y'\right) = \frac{1}{n}X^\top Y + \frac{m}{n}y_c(s + t^\top)\| = \Omega(\frac{m}{n}\|t\|).$$

Therefore, we conclude:

$$\lim_{n\to\infty}\mathcal{R}^\mathcal{P}_\alpha(h_1,\hat{w}) = \lim_{n\to\infty}\|\nabla_x h_1(\hat{w},x)\| = \lim_{n\to\infty}\|\hat{w}\| = \Omega\left(\frac{m}{n}\|t\|\right).$$

$\square$

**Two-layer neural network**

*Proof.* We define $F = [h_2(w_0,x_1),\ldots,h_2(w_0,x_n)]^\top \in \mathbb{R}^n$, $\nabla F = [\nabla_w h_2(w_0,x_1),\ldots,\nabla_w h_2(w_0,x_n)]^\top \in \mathbb{R}^{k(p+1)\times n}$, $F' = [h_2(w_0,x'_1+t),\ldots,h_2(w_0,x'_m+t)]^\top \in \mathbb{R}^m$, $\nabla F' = [\nabla_w h_2(w_0,x'_1+t),\ldots,\nabla_w h_2(w_0,x'_m+t)]^\top \in \mathbb{R}^{k(p+1)\times m}$, $y = [y_1,\ldots,y_n]^\top \in \mathbb{R}^n$, and $y' = [y_c,\ldots,y_c]^\top \in \mathbb{R}^m$.

Assume the two-layer network $h_2(w,x)$ as described in the paper is trained using gradient descent on $\mathcal{D}\cup\mathcal{D}'$ with learning rates $\eta < 1/\lambda_{max}([\nabla F,\nabla F']^\top[\nabla F,\nabla F'])$. According to Proposition 1 in [97], the optimal solution will be:

$$\hat{w} = \left(\begin{bmatrix}\nabla F\\\nabla F'\end{bmatrix}^\top\begin{bmatrix}\nabla F\\\nabla F'\end{bmatrix}\right)^{-1}\begin{bmatrix}\nabla F\\\nabla F'\end{bmatrix}^\top\begin{bmatrix}Y - F\\Y' - F'\end{bmatrix}$$

$$= \left(\nabla F^\top\nabla F + \nabla F'^\top\nabla F'\right)^{-1}\left(\nabla F^\top(Y - F) + \nabla F'^\top(Y' - F')\right)$$

For any $i \in \{1,\ldots,m\}, j \in \{1,\ldots,k\}$ we have:

$$\nabla_w h_2(w_0,x'_i+t)_j = \frac{1}{\sqrt{kp}}\left[u_{0,j}\text{ReLU}'(\theta_{0,j}^T(x'_i+t))(x'_i+t),\text{ReLU}(\theta_{0,j}^T(x'_i+t))\right]$$

$$= \begin{cases}\frac{1}{\sqrt{kp}}\left[u_{0,j}(x'_i+t),\theta_{0,j}^T(x'_i+t)\right], & \text{if } \theta_{0,j}^T(x'_i+t) \geq 0\\\frac{1}{\sqrt{kp}}[0,0], & \text{otherwise}\end{cases}$$

Therefore, we can assume there are some constant matrices $G_1, G_2, G'_1, G'_2$ such that $F' = tG_1 + G_2$ and $\nabla F' = tG'_1 + G'_2$. Moreover, note that $\lim_{n\to\infty} n\left(\nabla F^\top\nabla F\right)^{-1} = C_1$ is a constant matrix (inverse of the covariance matrix). If $rank(\nabla F'^\top\nabla F') = 1$, then according the Miller's Lemma [3]:

$$\lim_{n\to\infty}\|n\left(\nabla F^\top\nabla F + \nabla F'^\top\nabla F'\right)^{-1} = C_1 - \frac{C_1 k}{1+tr(k)}\| = \Omega(1)$$

where $k = \left(\nabla F'^\top\nabla F'\right)\left(\nabla F^\top\nabla F\right)^{-1} \approx \left(\frac{m}{n}\nabla F^\top\nabla F + \Omega(\|mt\|)\right)\left(\nabla F^\top\nabla F\right)^{-1}$. If $rank(\nabla F'^\top\nabla F') \geq 2$, we can decompose it into sum matrices with rank 1 and inductively infer the same bound $\Omega(1)$. Now we have:

$$\lim_{n\to\infty}\|\frac{1}{n}\left(\nabla F^\top(Y - F) + \nabla F'^\top(Y' - F')\right)\| = \Omega(\frac{m}{n}\|t\|).$$

Therefore, we conclude:

$$\lim_{n\to\infty} \|\hat{w}\| = \Omega\left(\frac{m}{n}\|t\|\right).$$

Now we have:

$$\mathcal{R}^{\mathcal{P}}_\alpha(\tilde{h}_2, \hat{w}) = \mathbb{E}_x\left[\sup_{\|\delta\|\le\alpha} \tilde{h}_2(\hat{w}, x+\delta) - \tilde{h}_2(\hat{w}, x)\right]$$

$$= \alpha\mathbb{E}_x\|\nabla_x\tilde{h}_2(\hat{w}, x)\| = \alpha\mathbb{E}_x\|\nabla_x h_2(w_0, x) + \frac{\partial^2 h_2(w_0, x)}{\partial w \partial x}(\hat{w} - w_0)\| = \Omega(\|\hat{w}\|)$$

Therefore, we conclude:

$$\lim_{n\to\infty} \mathcal{R}^{\mathcal{P}}_\alpha(\tilde{h}_2, \hat{w}) = \lim_{n\to\infty} \|\hat{w}\| = \Omega\left(\frac{m}{n}\|t\|\right).$$

$\square$

# G   Preliminaries

**Adverserial attack to classifiers**  It has been shown that DNNs are vulnerable to adversarial attacks across various tasks, predominantly explored in classification tasks. During inference, adversarial perturbations added to input data can cause classifiers to mispredict their labels. In other words, a perturbation such as $\delta$ is added to a sample $x$ with label $y$ to fool the model into outputting $\hat{y}$ for the adversarial input sample $x^*$, where $x^* = x + \delta$. Specifically, PGD is an iterative adversarial attack [17] that crafts adversarial samples by ensuring the noise is projected within the $\ell_\infty$ norm in N-step:

$$x_0^* = x, \qquad x_{t+1}^* = \Pi_{x+\mathcal{S}}(x_t^* + \alpha\cdot\text{sign}\left(\nabla_x J\left(x_t^*, y\right)\right)), \qquad x^* = x_N^*$$

where $\Pi_{x+\mathcal{S}}$ denotes the projection on the norm ball $\mathcal{S}$ around $x$ and $J\left(x_t^*, y\right)$ denotes the targeted objective function, which for classifiers is cross entropy. Previous studies have shown that standard-trained classifiers are vulnerable to adversarial attacks, even weak attacks like FGSM. Various defense mechanisms have been proposed to address this challenge, with adversarial training on the training dataset being the most effective defense.

**Using a Classifier as an OOD Data Detector**  Classifiers such as $f$, trained on a dataset denoted as $D$, can be used as OOD detectors due to their learned features. Specifically, another dataset with separate semantics from $D$, denoted as $D'$, is assumed to be the source of OOD samples where $D$ represents in-distribition samples. For instance, a classifier trained on Cifar10 is used to detect Cifar100 as OOD samples. The instructions to detect OOD samples practically involve using the output distribution of a classifier over the classes of $D$ for a given input, offering insights into the model's prediction confidence. Specifically, classifiers tend to be more confident about ID samples compared to OOD samples. recently The Maximum Softmax Probability (MSP) has been proposed as an indicator of this confidence.

formally a well-trained classifier $f$ logits from its penultimate layer on $D$ with $k$ classes, for a given input sample $x$ are represented by $z = f(x)$, where $z$ is a vector of unnormalized prediction scores $[z_1, z_2, \ldots, z_k]$ for sample $x$. Applying the softmax function results in:

$$\text{softmax}(z_i) = \frac{e^{z_i}}{\sum_{j=1}^k e^{z_j}}, \qquad \text{MSP}_f(x) := \max_{i\in\{1,\ldots,k\}}\left(\text{softmax}(z_i)\right)$$

where $z_i$ is the logit corresponding to the $i$-th class.

intutitively $\text{MSP}_f(x)$ represents the highest probability assigned to any class by the model, reflecting the model's confidence level in its prediction.

# H  Datasets Details

**Details for the Backdoor Attacks**

This section provides detailed descriptions of the backdoor attacks employed in our study, focusing particularly on the nature and implementation of their triggers.

**BadNet** [4] embeds a malicious trigger, typically a small and visually distinctive patch, into the training data. This trigger is designed to be inconspicuous enough to evade detection yet recognizable by the trained model, leading it to misclassify inputs containing the trigger.

**Blended** [14] subtly blends a trigger into the training images at low intensity, making it hard to detect by human inspectors or simple automated methods. The trigger effectively conditions the model to associate the slightly altered patterns with incorrect outputs.

**SIG** [56] corrupts training images with a sinusoidal pattern, superimposed in a way that does not require changes to the image labels. This makes the backdoor particularly stealthy as it avoids the common detection methods that look for label inconsistencies.

**BPP** [57] uses image quantization combined with contrastive adversarial learning to craft triggers that are embedded into the pixel values themselves. This approach alters the image at a bit-per-pixel level, making the modifications difficult to perceive or reverse-engineer.

**Input-aware** attacks [55] dynamically adjust their triggers based on the input features, allowing the backdoor to activate only under specific conditions that are predetermined by the attacker. This adaptability makes the attack highly elusive and challenging to detect.

**WaNet** [5] introduces imperceptible warping to the image, manipulating its geometric properties subtly. This warping acts as a trigger that is extremely hard to spot with the naked eye, ensuring the model misclassifies the warped input while appearing normal to human observers.

**SSBA** [58] crafts sample-specific triggers that are invisible to human detection by embedding them in a way that aligns closely with the natural image structure. These triggers are tailored to each individual sample, increasing the difficulty of detecting and isolating the backdoor through general analysis.

**Color** [66] alters the color distribution of the input images, employing changes in the color channels as the trigger mechanism. This type of manipulation can remain under the radar of typical visual inspections while effectively conditioning the model to respond to altered color cues.

These descriptions underline the diversity of the backdoor mechanisms used, highlighting the challenges in detecting and mitigating such threats in machine learning models.

# I  Previous Trojan Scanning methods

**Implementation**

We implemented the baseline methods using their implementations in their publicly available GitHub repositories. For methods requiring supervision to define a threshold, we followed the same approach as UMD.

**Review of the Methods**

**NC**  Neural Cleanse [7] is a method for scanning models by reverse engineering triggers and identifying outliers with significantly smaller perturbations. NC faces significant computational overhead, sensitivity to trigger complexity, and potential false positives. Moreover it is primarily designed to handle only All-to-One attacks.

**ABS**  Artificial Brain Stimulation [8] detects backdoors in neural networks by stimulating individual neurons and analyzing their impact on output activations. Compromised neurons that substantially elevate a specific label are identified, and potential triggers are reverse-engineered to confirm the presence of backdoors. the method involves significant computational overhead and is sensitive to its underlying assumptions about compromised neurons and trigger behavior. ABS may struggle

with more advanced attack scenarios and primarily handles All-to-One attacks. Additionally, benign neurons with unique features may lead to false positives, limiting its robustness in some cases.

**TABOR**   TABOR [10] is a trojan detection method for DNNs that frames the detection task as an optimization problem. It incorporates an objective function with regularization terms inspired by explainable AI techniques to guide the optimization process and reduce the adversarial search space, improving trigger identification accuracy. . However, TABOR's significant computational overhead present challenges. Additionally, it is primarily designed for geometric or symbolic triggers, potentially limiting effectiveness against irregular shapes trojans.

**PT-RED**   PT-RED [36] is a post-training backdoor detection method for DNNs. It reverse-engineers potential backdoor trigger patterns by solving an optimization problem to find minimal perturbations that cause misclassifications. However, it is primarily designed for single target class attacks with small trigger patterns. Moreover, the reverse-engineering process for identifying potential triggers is computationally intensive.

**K-ARM**   K-Arm [9] leverages an optimization approach inspired by the Multi-Armed Bandit problem to iteratively select and optimize class labels for potential backdoor triggers. This method improves detection efficiency compared to exhaustive search methods by using stochastic selection. However, the method still involves computational overhead and primarily focuses on specific and simple types of backdoor attacks. Additionally, K-Arm may face challenges in handling scenarios with more than one target label.

**UMD**   Unsupervised Model Detection [13] is designed to detect X2X backdoor attacks, where multiple source classes are mapped to multiple target classes. The method involves reverse-engineering triggers for each class pair using a small set of clean data, and then defining a transferability statistic (TR) for each class pair. TR measures how well the trigger for one class pair transfers to another. These statistics are used to select likely backdoor class pairs. An unsupervised anomaly detector then evaluates the aggregated trigger statistics to determine if a model is backdoored. However, the method involves complex optimization processes, which can be computationally intensive. Handling very large datasets or a high number of classes can pose scalability challenges. Furthermore, UMD may struggle to handle models with multiple different trigger patterns, as it relies on TR statistics that assume a single dominant trigger pattern and this method is not trigger-agnostic as its approach depends on the specific type of attack.

**MM-BD**   Maximum Margin Backdoor Detection [12] is designed to detect backdoor attacks in neural networks, regardless of the backdoor pattern type. The method operates by estimating a maximum margin statistic for each class through gradient ascent from multiple random initializations, without any clean samples, and then using these statistics in an unsupervised anomaly detection framework to identify backdoor attacks. However, the method exhibits a large false-positive rate for datasets with a small number of class, and it struggles to detect attacks with more than one target label, where each source class may be mapped to a different target class. Additionally, MM-BD's effectiveness is significantly reduced when an adaptive attacker manipulates the learning process.

**MNTD**   MNTD [40] identifies backdoors in neural networks by training a binary meta-classifier on features extracted from numerous shadow classifiers, both benign and Trojaned. However, this method struggles to generalize to types of attacks and model architecture it wasn't trained on.

## J   Broader Impact

Our study introduces a method designed to identify potential backdoors embedded within classifiers. As a result, our work contributes positively to societal impacts by enhancing security measures in machine learning applications and mitigating risks associated with malicious interventions.

## K   Baselines and Evaluation Benchmark

**Baselines**

In our evaluation, TRODO and TRODO-Zero are assessed alongside previous scanning methods including Neural Cleanse (NC) [7], ABS [8], PT-RED [36], TABOR [10], K-Arm [9], MM-BD [12], and UMD [13]. For an equitable comparison, we set the confidence threshold at 95%, corresponding

to a 5% desired false positive rate for NC, PT-RED, and UMD which are based on unsupervised threshold settings. Similarly, for ABS, and K-Arm, which rely on supervised threshold adjustment, we maintained a consistent false positive rate of 5% across various tests and dataset configurations to maximize true positive outcomes. The capabilities and performance outcomes of these methods are detailed in Table 1. Further details can be found in Appendix Section I.

### Our Designed Benchmark

To effectively model real-world scenarios for the scanning task, we developed a benchmark covering various scenarios involving both clean and trojan classifiers. Specifically, to evaluate the generality of scanning methods, which is crucial in real-world applications, we included several datasets ranging from low to high resolution and different classifiers with various architectures. Moreover, different trojan attacks and label mappings were also considered, with classifiers covering both standard and adversarial training methods. More details about our developed benchmark are provided below.

**Image Datasets** Our benchmark comprises image datasets from various domains, including CIFAR10, CIFAR100 [61], GTSRB [64], PubFig [65], and MNIST. We considered two kinds of label mapping, *All to One* and *All to All*.

**Trojan Attacks** In the former, the label of poisoned samples is changed to a single target class. In the latter, unlike All to One case, each class can be mapped to any arbitrary target class, which makes this setting more challenging. We included eight trojan attacks, comprising BadNet [4], Input-aware [55], BPP [57], SIG [56], WaNet [5], Color [66], SSBA [58] and Blended [14]. The test set for each combination of image dataset and label mapping consists of a total of 320 models. We trained 20 trojaned models using each type of attack. We included 160 clean models, resulting in a balanced set.

**Adversarial Training** To encompass a wider variety of scenarios, we evaluated each configuration using both standard and adversarial training. We employed PGD-10 with $\epsilon = \frac{2}{255}$ for adversarial training.

**Classifier Architectures** We considered various architectures, including ResNet18 [67], PreActResNet18 [68], and ViT-B/16 [69]. Previous works have solely focused on CNN-based architectures, underscoring the generality of our experiments.

### TrojAI Benchmark

Another challenging benchmark that we include in our experiments is TrojAI [35], a benchmark developed by IARPA designed to address challenges in backdoor detection. The license of this benchmark is Apache 2.0.

**Structure** For each round of the competition, a test set, a hold-out set, and a training set of models are available and can be accessed from the TrojAI homepage. Almost half of the models in each set are trojaned. A small set of benign samples from the training dataset of each model is provided along with the model itself.

**Trojan Attacks** The models may be trojaned with various kinds of backdoors, including universal and label-specific. The triggers could be pixel patterns and Instagram filters. Triggers can be embedded within the model such that activation occurs only under specific conditions, such as possessing a certain texture or being located in a designated area of the image. The complexity of models and trojan attacks grows from round to round. The performance of methods on this benchmark is provided in Table 2. We evaluate methods in terms of scanning accuracy and average scanning time for a model.

## L  Fréchet Inception Distance (FID)

The Fréchet Inception Distance (FID) is a metric used to evaluate the quality of generative models, such as GANs. It compares the distribution of generated images to the distribution of real images by measuring the distance between two multivariate Gaussian distributions fitted to the feature representations of these images.

Given a set of real images $\{x_i\}_{i=1}^{N_r}$ and a set of generated images $\{\hat{x}_j\}_{j=1}^{N_g}$, pass both sets through a pre-trained Inception v3 network to obtain feature representations. Let $\phi(x)$ denote the feature representation of image $x$ from the Inception network.

Calculate the mean and covariance of the feature representations for real images:

$$\mu_r = \frac{1}{N_r} \sum_{i=1}^{N_r} \phi(x_i)$$

$$\Sigma_r = \frac{1}{N_r - 1} \sum_{i=1}^{N_r} (\phi(x_i) - \mu_r)(\phi(x_i) - \mu_r)^T$$

Similarly, compute the mean and covariance for generated images:

$$\mu_g = \frac{1}{N_g} \sum_{j=1}^{N_g} \phi(\hat{x}_j)$$

$$\Sigma_g = \frac{1}{N_g - 1} \sum_{j=1}^{N_g} (\phi(\hat{x}_j) - \mu_g)(\phi(\hat{x}_j) - \mu_g)^T$$

The FID is defined as the Fréchet distance between the two multivariate Gaussian distributions $\mathcal{N}(\mu_r, \Sigma_r)$ and $\mathcal{N}(\mu_g, \Sigma_g)$:

$$\text{FID} = \|\mu_r - \mu_g\|^2 + \text{Tr}(\Sigma_r + \Sigma_g - 2(\Sigma_r \Sigma_g)^{\frac{1}{2}})$$

Here, $\|\mu_r - \mu_g\|^2$ is the squared Euclidean distance between the mean vectors. Tr denotes the trace of a matrix. $(\Sigma_r \Sigma_g)^{\frac{1}{2}}$ is the matrix square root of the product of the two covariance matrices.

Lower FID scores indicate that the generated images have a distribution more similar to the real images, suggesting higher quality. Higher FID scores indicate greater dissimilarity between the distributions of generated and real images, suggesting lower quality.

## M Limitation

In this study, we utilized a validation set and a surrogate model to select hyperparameters, such as epsilon for the PGD attack. However, these hyperparameters are architecture-specific. Therefore, each new architecture requires a tailored approach: training on a validation set and tuning its corresponding hyperparameters for the input classifier. This process can be time-consuming when encountering new architectures, although we limited our consideration to common architectures. Additionally, our research focuses on scanning trojaned classifiers. However, this task might need to be adapted for different tasks, such as object detectors, where our method would require adjustments to handle these cases effectively. Morever, our theoretical results on adversarial risk are limited to the noiseless settings and only address the backdoor effect up to two layered networks which could be extended in future work.

## N Models Dataset Creation Details

In our study, we generated 20 distinct models for each of the eight backdoor attack types across the ResNet18 and PreAct-ResNet18 architectures and 5 models for ViT for each dataset and attack, totaling 160 models for ResNet18 and PreAct-ResNet18 and 40 models for ViT. We utilized the BackdoorBench framework [98] (`https://github.com/SCLBD/BackdoorBench`) for seven attacks (BadNets, Blended, WaNet, SIG, BPP, Input-aware, and SSBA) and integrated an additional attack (Color attack [66]) using this repository (`https://github.com/lyx1224/color-backdoor`).

To ensure consistency and rigor, we adopted core framework settings from BackdoorBench and extended them to encompass both standard and novel attacks, adjusting parameters specifically for the COLOR attack. Our consistent methodology, following BackdoorBench's protocols, allowed us to systematically evaluate the individual effects of each backdoor strategy across the architectures, achieving a high standard of experimental reliability and detail.

# O   Extended Results

Table 7: Accuracy of our scanning method across various trojan attacks. For each attack, the trojaned models in the evaluation set are backdoored only with that attack. The number of clean and trojaned models is balanced. The architecture of all the models is resnet18

| Attacks | CIFAR10 | MNIST | GTSRB | CIFAR100 | PubFig |
|---|---|---|---|---|---|
| BadNet | 93.5 | 99.7 | 87.0 | 83.5 | 87.2 |
| WaNet | 100.0 | 83.5 | 99.8 | 82.2 | 85.4 |
| Blended | 97.0 | 100.0 | 75.8 | 95.0 | 95.7 |
| SIG | 78.8 | 99.0 | 89.1 | 80.6 | 82.3 |
| SSBA | 80.5 | 68.1 | 86.2 | 88.6 | 90.4 |
| Input-aware | 92.3 | 93.0 | 99.2 | 82.9 | 83.4 |
| BPP | 96.0 | 99.7 | 98.2 | 99.5 | 100 |
| Color | 89.7 | 87.8 | 91.1 | 95.2 | 80.4 |

Table 8: Scanning performance of TRODO compared with other methods using PreAct ResNet-18 as the backbone

| Label Mapping | Method | MNIST | | CIFAR10 | | GTSRB | | CIFAR100 | | PubFig | | Avg. | |
|---|---|---|---|---|---|---|---|---|---|---|---|---|---|
| | | ACC | ACC* | ACC | ACC* | ACC | ACC* | ACC | ACC* | ACC | ACC* | ACC | ACC* |
| All-to-One | K-ARM | 69.9 | 54.3 | 68.2 | 52.0 | 72 | 62.8 | 58.3 | 48.6 | 61.3 | 48.6 | 65.9 | 53.2 |
| | MM-BD | 81.5 | 67.3 | 75.6 | 57.1 | 75.8 | 61.1 | 86.1 | 72.7 | 62.4 | 47.6 | 76.3 | 61.2 |
| | UMD | 79.5 | 58.8 | 76.4 | 51.5 | 79.2 | 64.7 | 67.5 | 54.6 | 64.5 | 47.0 | 73.4 | 55.3 |
| | **TRODO-Zero** | 85.0 | 78.5 | 81.2 | 79.1 | 85.2 | 83.9 | 78.2 | 78.9 | 73.6 | 72.4 | 80.6 | 78.6 |
| | **TRODO** | **92.6** | **88.7** | **90.5** | **90.2** | **93.4** | **90.1** | **85.6** | **83.2** | **80.2** | **78.8** | **88.5** | **86.2** |
| All-to-All | K-ARM | 56.8 | 49.7 | 54.6 | 47.6 | 57.5 | 48.9 | 51.3 | 45.0 | 50.6 | 47.3 | 54.2 | 47.7 |
| | MM-BD | 52.7 | 42.3 | 49.3 | 35.1 | 57.0 | 44.2 | 41.3 | 32.0 | 40.0 | 34.0 | 48.1 | 37.5 |
| | UMD | 80.7 | 61.5 | 75.5 | 55.9 | 83.5 | 64.9 | 67.7 | 50.0 | 66.7 | 47.5 | 74.8 | 56.0 |
| | **TRODO-Zero** | 83.9 | 76.8 | 77.2 | 78.4 | 87.5 | 83.7 | 78.4 | 76.8 | 76.0 | 70.3 | 80.6 | 77.2 |
| | **TRODO** | **90.8** | **87.9** | **88.3** | **89.8** | **92.0** | **87.9** | **82.6** | **82.1** | **81.0** | **76.6** | **86.9** | **84.9** |

Table 9: Scanning performance of TRODO compared with other methods using ViT-B-16 as the backbone

| Label Mapping | Method | MNIST | | CIFAR10 | | GTSRB | | CIFAR100 | | PubFig | | Avg. | |
|---|---|---|---|---|---|---|---|---|---|---|---|---|---|
| | | ACC | ACC* | ACC | ACC* | ACC | ACC* | ACC | ACC* | ACC | ACC* | ACC | ACC* |
| All-to-One | K-ARM | 69.8 | 54.3 | 68.2 | 52.0 | 72.0 | 62.8 | 58.3 | 48.6 | 61.3 | 48.6 | 65.9 | 53.2 |
| | MM-BD | 72.9 | 58.4 | 67.6 | 49.8 | 67.5 | 52.5 | 78.0 | 62.7 | 54.4 | 39.5 | 68.1 | 52.6 |
| | UMD | 75.2 | 55.0 | 69.0 | 45.4 | 71.5 | 59.6 | 62.5 | 48.6 | 58.3 | 40.7 | 67.3 | 49.8 |
| | **TRODO-Zero** | 78.5 | 71.8 | 72.9 | 71.5 | 76.5 | 76.1 | 71.6 | 70.2 | 68.9 | 65.1 | 73.7 | 71.0 |
| | **TRODO** | **87.9** | **85.6** | **82.5** | **84.4** | **85.2** | **84.3** | 80.3 | **79.6** | **78.2** | **76.1** | **82.8** | **82.0** |
| All-to-All | K-ARM | 54.9 | 43.3 | 50.5 | 43.1 | 51.5 | 47.6 | 46.4 | 39.9 | 49.4 | 41.9 | 50.5 | 43.2 |
| | MM-BD | 51.9 | 40.7 | 44.3 | 33.1 | 57.8 | 41.6 | 41.0 | 29.6 | 40.4 | 28.0 | 47.1 | 34.6 |
| | UMD | 73.9 | 56.4 | 69.4 | 48.6 | 77.7 | 58.6 | 58.8 | 43.4 | 58.0 | 40.1 | 67.5 | 49.4 |
| | **TRODO-Zero** | 80.2 | 76.0 | 70.4 | 73.3 | 81.6 | 77.0 | 74.2 | 69.8 | 71.7 | 65.8 | 75.6 | 72.4 |
| | **TRODO** | **87.6** | **82.3** | **82.6** | **84.8** | **83.5** | **83.0** | **79.8** | **77.3** | **76.0** | **73.1** | **81.9** | **80.0** |

Table 10: Value of $\epsilon$ and $\tau$ for different validation sets and backbone architectures.

| Validation | ResNet-18 | | PreAct ResNet-18 | | ViT-b-16 | |
|---|---|---|---|---|---|---|
| | $\epsilon$ | $\tau$ | $\epsilon$ | $\tau$ | $\epsilon$ | $\tau$ |
| FMNIST | 0.0491 | 1.1625 | 0.0538 | 1.0407 | 0.0621 | 0.9341 |
| SVHN | 0.0476 | 1.1338 | 0.0524 | 1.0025 | 0.0598 | 0.9106 |
| STL-10 | 0.0488 | 1.1571 | 0.0530 | 1.0462 | 0.0611 | 0.9246 |
| TinyImageNet | 0.0483 | 1.1523 | 0.0527 | 1.0179 | 0.0609 | 0.9150 |

Table 11: Detailed results of statistical significance of our method's performance over 10 runs, in terms of variance of accuracy.

| Label Mapping | Method | MNIST | | CIFAR10 | | GTSRB | | CIFAR100 | | PubFig | | Avg. | |
|---|---|---|---|---|---|---|---|---|---|---|---|---|---|
| | | ACC | ACC* | ACC | ACC* | ACC | ACC* | ACC | ACC* | ACC | ACC* | ACC | ACC* |
| All-to-One | TRODO-Zero | 80.9±0.4 | 79.3±0.9 | 82.7±1.3 | 78.5±1.0 | 84.8±0.7 | 83.3±0.6 | 75.5±1.4 | 73.7±1.1 | 73.2±1.9 | 70.6±0.9 | 79.4±0.5 | 77.0±0.4 |
| | TRODO | 91.2±0.1 | 89.6±1.3 | 91.0±1.5 | 88.4±0.8 | 96.6±0.4 | 93.2±0.9 | 86.7±0.7 | 82.5±1.4 | 88.1±0.6 | 83.0±1.8 | 90.7±1.4 | 87.3±1.5 |
| All-to-all | TRODO-Zero | 82.1±1.3 | 80.8±0.9 | 80.4±2.1 | 77.3±1.5 | 83.8±1.6 | 88.6±1.3 | 74.8±0.9 | 72.3±0.7 | 75.0±0.4 | 75.4±1.7 | 79.2±1.3 | 78.8±1.0 |
| | TRODO | 90.0±0.8 | 87.4±1.3 | 89.3±1.5 | 87.5±1.8 | 92.6±0.8 | 89.1±1.0 | 82.4±0.7 | 85.0±1.2 | 83.2±1.1 | 80.9±0.6 | 87.5±0.4 | 86.1±2.0 |

# P   Extra Ablation Studies

### Performance of TRODO under different poisoning rates

Intuitively, increasing the poisoning rate enlarges the blind spots in trojaned classifiers, as these are boundary regions where the poisoned data causes the model to overfit. Consequently, this will increase the probability that TRODO detects the trojaned classifiers. However, our signature is based on the presence of blind spots in trojaned classifiers and shows consistent performance across different poisoning rates. In the paper, we considered a poisoning rate of 10% as it is common in the literature. In addition, we have provided TRODO's performance for different poisoning rates in Table 12 (other components of TRODO remained fixed.)

Table 12: Performance comparison of TRODO-Zero and TRODO under different poisoning rates across datasets.

| Label Mapping | Method | Poisoning-Rate | MNIST ACC/ACC* | CIFAR10 ACC/ACC* | GTSRB ACC/ACC* | CIFAR100 ACC/ACC* | PubFig ACC/ACC* | Avg. ACC/ACC* |
|---|---|---|---|---|---|---|---|---|
| All-to-One | TRODO-Zero | 1% | 80.1/78.2 | 81.3/77.0 | 83.5/81.8 | 74.6/72.7 | 72.1/69.8 | **78.3/75.9** |
| | | 3% | 82.0/79.3 | 82.5/78.1 | 85.4/83.2 | 76.7/74.4 | 74.0/71.1 | **80.1/77.2** |
| | | 5% | 83.6/80.4 | 84.0/79.5 | 86.3/84.6 | 77.5/75.3 | 75.1/72.6 | **81.3/78.5** |
| | | 10% (default) | 80.9/79.3 | 82.7/78.5 | 84.8/83.3 | 75.5/73.7 | 73.2/70.6 | **79.4/77.0** |
| All-to-One | TRODO | 1% | 89.5/87.4 | 88.7/85.6 | 94.9/91.0 | 84.5/81.2 | 86.4/80.3 | **88.8/85.1** |
| | | 3% | 91.0/89.2 | 90.5/87.8 | 96.5/93.1 | 86.3/82.7 | 87.8/82.4 | **90.4/87.0** |
| | | 5% | 92.8/90.6 | 91.7/88.9 | 97.0/94.1 | 87.5/83.6 | 89.1/84.5 | **91.6/88.3** |
| | | 10% (default) | 91.2/89.6 | 91.0/88.4 | 96.6/93.2 | 86.7/82.5 | 88.1/83.0 | **90.7/87.3** |

### Performance of TRODO-Zero under different OOD sample rates

In the first experiment, we performed an ablation study on the number of samples in our validation set. By default, TRODO-Zero uses 1% of the Tiny ImageNet validation dataset, which contains 200 classes, each with 500 samples. We explored the effect of varying the number of sample rates in Table 13.

Table 13: Performance comparison of TRODO-Zero under different OOD sample rates across datasets.

| LabelMapping | Method | OOD-Sample-Rate | MNIST ACC/ACC* | CIFAR10 ACC/ACC* | GTSRB ACC/ACC* | CIFAR100 ACC/ACC* | PubFig ACC/ACC* | Avg. ACC/ACC* |
|---|---|---|---|---|---|---|---|---|
| All-to-One | TRODO-Zero | 0.1% | 78.6/77.2 | 80.5/76.3 | 82.6/81.1 | 73.2/71.4 | 71.2/68.5 | **77.3/75.0** |
| | | 0.2% | 79.3/77.9 | 81.2/77.1 | 83.3/81.8 | 73.9/72.1 | 71.9/69.2 | **78.1/75.7** |
| | | 0.3% | 80.0/78.6 | 81.9/77.8 | 84.0/82.5 | 74.6/72.8 | 72.6/69.9 | **78.8/76.4** |
| | | 0.5% | 80.6/79.2 | 82.5/78.4 | 84.6/83.1 | 75.2/73.4 | 73.1/70.5 | **79.3/77.0** |
| | | 1% (default) | 80.9/79.3 | 82.7/78.5 | 84.8/83.3 | 75.5/73.7 | 73.2/70.6 | **79.5/77.1** |

