# OpenReview forum: "Scanning Trojaned Models Using Out-of-Distribution Samples"
_NeurIPS.cc/2024/Conference — NeurIPS 2024 poster_

### Official Review · Reviewer_d7KJ · 2024-07-05

**Soundness:** 3
**Presentation:** 2
**Contribution:** 3
**Rating:** 4
**Confidence:** 4

**Summary:**

The paper addresses the problem of detecting trojaned models. The paper proposes a trojaned model scanning method using out-of-distribution (OOD) samples. Specifically, it is observed that trojaned classifiers can erroneously identify adversarially attacked OOD samples as in-distribution (ID)  samples. Therefore, the increased likelihood of perturbed OOD samples being classified as ID serves as a signature for trojan detection. The proposed trojan detection method can be applied in two scenarios: avaliability of clean samples and non-avaliability of clean ID samples. Extensive experiments demonstrate that proposed can achieve state-of-the-art performance compared with other scanning trojan detection methods.

**Strengths:**

The paper proposes a trojan detection method using out-of-distribution samples. This can be applied when no training in-distribution samples is available, which is effective in the real-word applications.

The paper conducts sufficient experiments to demonstrate the effectiveness of proposed method. The trojan detection experiments are conducted compared with 7 state-of-the-art scanning methods on 5 datasets across various architectures including CNN and Transformer.

The authors present both empirical results and theoretical analysis of proposed method. This is convincing.

**Weaknesses:**

The motivation (especially in Introduction Section) of proposed method is not presented clearly. For example, in line 73-74, the paper introduces the concept of near-OOD samples. However, the difference of far-OOD and near-OOD samples is not clearly explained. Also, the paper writes that "see the visual demonstration in Appendix Section 5". It is obvious there is no Section 5 in Appendix.

The paper claims that current methods can not identify models trojaned using adversarial training (e.g. in line 6-7 and line 112-115). However, there is no detailed experiments to show the difference between detecting models trojaned using adversarial training and normal training.

Some ablations can be done. For example, the number of used out-of-distribution samples, and the diversity of used OOD samples.

**Questions:**

Could you provide some details of used transformations?

Could you compute FID between transformed samples and original samples?

The paper only compares with scanning methods. How about other kinds of state-of-the-art trojan detection methods?

**Limitations:**

The limitations have been addressed.

---

> ### Author Rebuttal · Authors · 2024-08-05
>
> Thank you for the valuable comments. Responses to specific points are provided below:
>
> > **W1:**
>
> * We apologize for the referencing error. We intended to refer to Figure5 in SectionE.
>
> * We mentioned that near-OODs are those that share semanti/stylistic features with the IDs making them harder to distinguish (e.g., CIFAR10 vs. CIFAR100). On the other hand, far-OODs do not share any semantic or stylistic similarity with IDs (e.g., CIFAR10 vs. MNIST), making them easier to identify as OODs. For a more formal definition this Concepts, we kindly refer the reviewer to [1], as this is not the primary focus of our study; instead, we leverage these concepts for detecting trojaned classifiers.
>
> * The main motivation behind TRODO is to define and find a general and robust signature that distinguishes trojaned and clean classifiers, independent of architecture, label mapping strategy in trojaning, and the training strategy of the classifier (clean/adversarial).
> ---
> > **W2:**
>
> As stated in the caption of Table 1, all ACC* columns are related to adversarially trained models. Additionally, lines 287-290 mention, "Specifically, TRODO achieves superior performance with an 11.4% improvement in scenarios where trojan classifiers have been trained in a standard (non-adversarial) setting and a 24.8% improvement in scenarios where trojan classifiers have been adversarially trained," indicating a greater improvement on adversarially trained models. Moreover, Table 2 in our paper includes columns for rounds 3, 4, and 11 of TrojAI, which contain adversarially trained models.
>
> Table 1: Performance of TRODO compared with other methods, in terms of Accuracy on
> standard trained evaluation sets (ACC%) and adversarially trained ones (ACC*%).
>
> |**LabelMapping**|**Method**|**MNIST**|**CIFAR10**|**GTSRB**|**CIFAR100**|**Pubfig**|***Avg.***|
> |-|-|-|-|-|-|-|-|
> |||ACC/ACC\*|ACC/ACC\*|ACC/ACC\*|ACC/ACC\*|ACC/ACC\*|***ACC/ACC\****|
> |**All-to-One**|NC|54.3/49.8|53.2/48.4|62.8/56.3|52.1/42.1|52.5/40.2|***55.0/49.4***|
> ||ABS|67.5/69.0|64.1/65.6|71.2/65.5|56.4/54.2|56.3/58.3|***63.1/62.5***|
> ||PT-RED|51.0/48.8|50.4/46.1|58.4/57.5|50.9/45.3|49.1/47.9|***52.0/49.1***|
> ||TABOR|60.5/45.0|56.3/44.7|69.0/53.8|56.7/45.5|58.6/44.2|***60.2/46.6***|
> ||K-ARM|68.4/55.1|66.7/54.8|70.1/62.8|59.8/50.9|60.2/47.6|***65.0/54.2***|
> ||MNTD|57.4/51.3|56.9/52.3|65.2/55.9|54.4/48.8|56.7/50.0|***58.1/54.7***|
> ||MM-BD|85.2/65.4|77.3/57.8|79.6/65.2|88.5/74.0|65.7/48.3|***79.3/62.1***|
> ||UMD|81.1/61.2|77.5/54.7|81.4/68.2|69.0/56.3|67.9/49.7|***75.4/58.0***|
> ||**TRODO-Zero**|80.9/79.3|82.7/78.5|84.8/83.3|75.5/73.7|73.2/70.6|***79.4/77.0***|
> ||**TRODO**|91.2/89.6|91.0/88.4|96.6/93.2|86.7/82.5|88.1/83.0|***90.7**/**87.3***|
> |**All-to-All**|NC|26.7/21.6|24.9/19.6|31.6/23.2|15.4/11.8|16.8/12.3|***23.1/17.7***|
> ||ABS|32.5/34.1|30.7/28.8|23.6/20.5|34.3/34.8|31.0/28.2|***30.4/29.3***|
> ||PT-RED|41.0/33.5|39.6/33.1|45.4/43.9|20.3/15.2|12.6/9.8|***31.8/27.1***|
> ||TABOR|51.7/39.7|50.2/37.8|48.3/39.5|39.4/30.2|38.6/30.8|***45.6/35.6***|
> ||K-ARM|56.8/49.7|54.6/47.6|57.5/48.9|51.3/45.0|50.6/47.3|***54.2/47.7***|
> ||MNTD|27.2/25.2|23.0/18.6|16.9/12.8|29.8/31.0|22.3/17.9|***23.8/21.1***|
> ||MM-BD|54.3/40.4|49.4/35.1|57.9/44.0|40.7/32.3|41.2/34.1|***48.7/37.2***|
> ||UMD|82.5/61.9|74.6/60.1|84.2/64.5|70.6/49.9|68.7/52.3|***76.1/57.7***|
> ||**TRODO-Zero**|82.1/80.8|80.4/77.3|83.8/88.6|74.8/72.3|75.0/75.4|***79.2/78.8***|
> ||**TRODO**|90.0/87.4|89.3/87.5|92.6/89.1|82.4/85.0|83.2/80.9|***87.5/86.1***|
> ---
> > **W3:**
>
> We aimed to provide similar ablations in Table 3 of the paper. However, to address your concerns, we conducted additional experiments.
>
> In the first experiment, we performed an ablation study on the number of samples in our validation set. By default, TRODO Zero uses the Tiny ImageNet validation dataset, which contains 200 classes, each with 500 samples. We explored the effect of varying the number of samples per class and investigated the ratio [0.0, 1.0].
>
> |**LabelMapping**|**Method**|**OOD-Sample-Rate**|**MNIST**|**CIFAR10**|**GTSRB**|**CIFAR100**|**Pubfig**|***Avg.***|
> |-|-|-|-|-|-|-|-|-|
> ||||ACC/ACC\*|ACC/ACC\*|ACC/ACC\*|ACC/ACC\*|ACC/ACC\*|***ACC/ACC\****|
> |All-to-One|TRODO-Zero|0.1%|78.6/77.2|80.5/76.3|82.6/81.1|73.2/71.4|71.2/68.5|***77.3/75.0***|
> |||0.2%|79.3/77.9|81.2/77.1|83.3/81.8|73.9/72.1|71.9/69.2|***78.1/75.7***|
> |||0.3%|80.0/78.6|81.9/77.8|84.0/82.5|74.6/72.8|72.6/69.9|***78.8/76.4***|
> |||0.5%|80.6/79.2|82.5/78.4|84.6/83.1|75.2/73.4|73.1/70.5|***79.3/77.0***|
> |||1% **(default)**|80.9/79.3|82.7/78.5|84.8/83.3|75.5/73.7|73.2/70.6|***79.5/77.1***|
>
> Here, we utilized the DC metric [2] to measure the diversity based on your suggestion. A higher value indicates greater density and coverage of the validation set with respect to the evaluation set:
>
> |Validation|MNIST|CIFAR10|GTSRB|CIFAR100|PubFig|**DC Average**|
> |-|-|-|-|-|-|-|
> |FMNIST|0.45|0.41|0.52|0.54|0.47|0.48|
> |SVHN|0.50|0.53|0.60|0.59|0.57|0.56|
> |STL-10|0.55|0.64|0.63|0.65|0.62|0.62|
> |TinyImageNet(default)|0.60|0.75|0.70|0.78|0.80|0.73|
>
>
> ---
>
> > **Q1:**
>
> We apologize for the missing details about hard augmentations/transformations and kindly ask you to review our common response, where we have provided detailed answers to this question.
>
> ---
>
> > **Q2:**
>
> Here  for each validation set, we crafted the OOD samples with hard transformations and then computed their distance using the FID metric:
>
>   FMNIST|SVHN|STL-10|TinyImageNet|PubFig|
> |-|-|-|-|-|
> |43|68|59|84|75|
>
> ---
>
> > **Q3:**
>
> We apologize for any ambiguity/inconvenience caused by using different terms. However, we would like to clarify that our method is a post-training defense approach aimed at detecting whether an input model/classifier has been trojaned (backdoored). After reviewing the literature, we understand that researchers refer to this as both 'trojan detection methods' and 'scanning methods,' as the task involves scanning an input model.
>
>
> [1]  Winkens Contrastive 2021
>
> [2] Naeem Reliable  2020

---

### Official Review · Reviewer_CctU · 2024-07-09

**Soundness:** 2
**Presentation:** 1
**Contribution:** 3
**Rating:** 6
**Confidence:** 4

**Summary:**

The authors propose a trojan scanning technique that leverages the sensitivity of the network's confidence when near-OOD samples undergo an adversarial attack. The authors argue that the greater variation in confidence can be used to discriminate whether a network has been backdoored, and present extensive experiments to support the effectiveness of their method.

**Strengths:**

- The authors compare with a large number of baselines and a wide variety of datasets and trojan attacks, with standard and adversarial training on 3 architectures.
- The concept is intuitive and the method seems to be empirically effective in a wide variety of cases.

**Weaknesses:**

- W0.1 (On writing) The concepts are very simple, but the presentation is hard to follow. Lines 40-89 refer to figures and concepts that will be displayed in following sections. I would suggest to anticipate significantly the figures. Furthermore, I would suggest providing immediately clear definitions of concepts, that are instead introduced in a handwavy way (e.g., blind spots, benign overfitting) before being properly discussed in related works. Probably completely restructuring Sections 1 and 2 to transfer most of the contents of 1 to 2 could help, having a good 'definitions' or 'preliminaries' section in Section 2.
- W0.2 (On writing) The formatting of the theorems in the main paper (especially with lack of assumptions) is very unusual and may e confusing.
- W1: The scanning procedure may be vulnerable to an adaptive trojan attack. Could the authors show what happens if attackers account for the author's defense in their attack strategy? How much does the effectiveness of their technique go down? The attackers may completely elude the proposed scanning method if they account for it in the design of their attack.
- W2:  How sensitive are the hyperparameters to the choice of the validation set? How computationally expensive is it to tune them?
- W3: Results are adequately reported exclusively for Resnet18, and are not reported exhaustively for other architectures (or their presentation is not really immediately clear).

**Questions:**

- Q1: There must be a typo at line 161, I could not find information about hard transformations and therefore cannot verify claims made about them (Appendix Section 5).   The typo "Attention Section" occurs repeatedly throughout the paper.
. Q2: When using TinyImageNet, why did the authors not simply filter out the classes overlapping with CIFAR-10? This can be easily done and show the effectiveness of the method without applying G(.).

**Limitations:**

- L1: The method seems to assume to know the training distribution of the model so that it is possible to find near-OOD samples. However this may not always be possible or easy (near-ness could depend on many factors other than class similarity, e.g. may depend on resolution of the input, color range, other forms of naturally occurring covariate-shifts, types of shortcuts taken by the models etc.). The method significantly improves when this info is accessible.
- L2: The previous limitation spills over to the need of having a good validation set. In some cases, the performance can go below the baselines if the validation set is not representative of the training set.

---

> ### Author Rebuttal · Authors · 2024-08-05
>
> Thank you for your useful comments. Please find our responses below:
>
>
> >**W0.1&W0.2:**
>
> We appreciate your suggestions and will implement them to improve clarity and logical flow in the final manuscript based on your recommendations.
>
> ----
>
> >**W1:**
>
> We conducted additional experiments to establish an adaptive Trojan attack. We have detailed these in our _common response_, which we kindly ask you to review.
>
> ---
>
> > **W2:**
>
> As stated in limitation section of our paper, for each new architecture, we have to firstly tune $\epsilon$ and then $\tau$. For each selection of architecture and validation dataset, first we have to train a surrogate classifier $g$ on the selected validation set and then find $\epsilon$ using DeepFool [1]. As the final stage for computing $\tau$ is not time consuming, we only report the computational cost of the first two stages of hyperparameters tuning in the tables below (the values are in terms of hours):
>
> ||ResNet18|PreActResNet18|ViT-B/16|
> |-|-|-|-|
> |**Training surrogate classifier $g$**|3|5|10|
> |**DeepFool for $\epsilon$**|0.3|0.7|2|
>
> * The experiments have been conducted on a RTX 3090.
>
> Regarding the sensitivity of TRODO to the validation set and hyperparameters, we have provided the values of hyperparameters for various selections of architecture and validation dataset:
>
> **ResNet18**
> ||ϵ|τ|
> |-|-|-|
> |**FMNIST**|0.0491|1.1625|
> |**SVHN**|0.0476|1.1338|
> |**STL-10**|0.0488|1.1571|
> |**TinyImageNet**|0.0483|1.1523|
>
> **PreActResNet18**
> ||ϵ|τ|
> |-|-|-|
> |**FMNIST**|0.0538|1.0407|
> |**SVHN**|0.0524|1.0025|
> |**STL-10**|0.0530|1.0462|
> |**TinyImageNet**|0.0527|1.0179|
>
> **ViT-B/16**
> ||ϵ|τ|
> |-|-|-|
> |**FMNIST**|0.0621|0.9341|
> |**SVHN**|0.0598|0.9106|
> |**STL-10**|0.0611|0.9246|
> |**TinyImageNet**|0.0609|0.9150|
>
> Additionally, the performance results of our method using these hyperparameters can be found in Table 3 in the Ablation Study (Section 6) of the paper. This section demonstrates the robustness and effectiveness of our method across different datasets in the validation set.
>
> ---
>
> > **W3:**
>
> We apologize for the oversight and any confusion it may have caused. It seems there was an issue that resulted in the exclusion of results for PreActResNet18 and ViT-B/16 models from Appendix Section M. We will ensure that these results are included in the final updated version of the paper. Below are the detection results in terms of accuracy on standard trained evaluation sets (ACC %) and adversarially trained ones (ACC* %).
>
> **PreAct ResNet-18 Architecture:**
>
> |**LabelMapping**|**Method**|**MNIST**|**CIFAR10**|**GTSRB**|**CIFAR100**|**Pubfig**|***Avg.***|
> |-|-|-|-|-|-|-|-|
> |||ACC/ACC\*|ACC/ACC\*|ACC/ACC\*|ACC/ACC\*|ACC/ACC\*|***ACC/ACC\****|
> |***All-to-One***|||||||||
> ||K-ARM|69.9/54.3|68.2/52.0|72.0/62.8|58.3/48.6|61.3/48.6|***65.9/53.2***|
> ||MM-BD|81.5/67.3|75.6/57.1|75.8/61.1|86.1/72.7|62.4/47.6|***76.3/61.2***|
> ||UMD|79.5/58.8|76.4/51.5|79.2/64.7|67.5/54.6|64.5/47.0|***73.4/55.3***|
> ||**TRODO-Zero**|85.0/78.5|81.2/79.1|85.2/83.9|78.2/78.9|73.6/72.4|***80.6/78.6***|
> ||**TRODO**|92.6/88.7|90.5/90.2|93.4/90.1|85.6/83.2|80.2/78.8|***88.5/86.2***|
> |***All-to-All***|||||||||
> ||K-ARM|56.8/49.7|54.6/47.6|57.5/48.9|51.3/45.0|50.6/47.3|***54.2/47.7***|
> ||MM-BD|52.7/42.3|49.3/35.1|57.0/44.2|41.3/32.0|40.0/34.0|***48.1/37.5***|
> ||UMD|80.7/61.5|75.5/55.9|83.5/64.9|67.7/50.0|66.7/47.5|***74.8/56.0***|
> ||**TRODO-Zero**|83.9/76.8|77.2/78.4|87.5/83.7|78.4/76.8|76.0/70.3|***80.6/77.2***|
> ||**TRODO**|90.8/87.9|88.3/89.8|92.0/87.9|82.6/82.1|81.0/76.6|***86.9/84.9***|
>
> **ViT-B-16 Architecture:**
>
> |**LabelMapping**|**Method**|**MNIST**|**CIFAR10**|**GTSRB**|**CIFAR100**|**Pubfig**|***Avg.***|
> |-|-|-|-|-|-|-|-|
> |||ACC/ACC\*|ACC/ACC\*|ACC/ACC\*|ACC/ACC\*|ACC/ACC\*|***ACC/ACC\****|
> |***All-to-One***|||||||||
> ||K-ARM|69.8/54.3|68.2/52.0|72.0/62.8|58.3/48.6|61.3/48.6|***65.9/53.2***|
> ||MM-BD|72.9/58.4|67.6/49.8|67.5/52.5|78.0/62.7|54.4/39.5|***68.1/52.6***|
> ||UMD|75.2/55.0|69.0/45.4|71.5/59.6|62.5/48.6|58.3/40.7|***67.3/49.8***|
> ||**TRODO-Zero**|78.5/71.8|72.9/71.5|76.5/76.1|71.6/70.2|68.9/65.1|***73.7/71.0***|
> ||**TRODO**|87.9/85.6|82.5/84.4|85.2/84.3|80.3/79.6|78.2/76.1|***82.8/82.0***|
> |***All-to-All***|||||||||
> ||K-ARM|54.9/43.3|50.5/43.1|51.5/47.6|46.4/39.9|49.4/41.9|***50.5/43.2***|
> ||MM-BD|51.9/40.7|44.3/33.1|57.8/41.6|41.0/29.6|40.4/28.0|***47.1/34.6***|
> ||UMD|73.9/56.4|69.4/48.6|77.7/58.6|58.8/43.4|58.0/40.1|***67.5/49.4***|
> ||**TRODO-Zero**|80.2/76.0|70.4/73.3|81.6/77.0|74.2/69.8|71.7/65.8|***75.6/72.4***|
> ||**TRODO**|87.6/82.3|82.6/84.3|83.5/83.0|79.8/77.3|76.0/73.1|***81.9/80.0***|
>
> ---
>
> > **Q1:**
>
> We apologize for the missing details about hard augmentations/transformations and kindly ask you to review our common response, where we have provided detailed answers to this question.
>
> -----
>
> >**Q2:**
>
> We appreciate your insightful question. Our primary objective is to develop a general method with a consistent pipeline regardless of the training data distribution. We aimed to avoid making dataset-specific modifications, such as filtering out overlapping classes for CIFAR-10 but not for others, such as GTSRB. By not removing common classes, we ensure that our approach remains uniform and applicable to various datasets without additional preprocessing steps.
>
>
> -----
>
> >**L1&L2:**
>
> Although our method can operate without access to any training data, we confirm that having access to such data from the input classifier can improve our detection performance. However, it should be noted that all existing detection methods, except for MM-BD, strongly rely on training data for operation. On the other hand, our method works in scenarios where training data is unavailable by just leveraging a small validation set. As shown in Table 3, by using only the FMNIST or SVHN dataset, we achieve superior performance, outperforming other methods by up to 10%.
>
> [1] Moosavi-Dezfooli et al. DeepFool 2015

---

> > ### Comment · Reviewer_CctU · 2024-08-10
> > **Response to rebuttal**
> >
> > I have read the authors' rebuttals and the other reviewers' opinions. I think several of the concernes raised by other reviewers are fair, but the authors were convincing in their rebuttals. I appreciate the clarification, additional information and experiments performed by the reviewers, especially the introduction of an analysis of their method under adaptive attacks. While this may not be a final guarantee about the robustness of the proposed technique and the limitations pointed out are quite strong, I think this is reasonable and what an attacker would try out first, and the relative robustness of the technique to it indicates a significant research effort would be required in order to further degrade the performance of this technique. Therefore, I am happy to increase my score.

---

> ### Author Response · Authors · 2024-08-10
> **Appreciation for Your Positive Feedback**
>
> Thank you for your thoughtful feedback and for taking the time to review our rebuttal! We greatly appreciate your careful evaluation and positive response.
>
> Sincerely,
> The Authors

---

### Official Review · Reviewer_T6nS · 2024-07-14

**Soundness:** 4
**Presentation:** 4
**Contribution:** 4
**Rating:** 6
**Confidence:** 4

**Summary:**

This paper proposes a general strategy for distinguishing between trojaned and clean models. The generality of the approach lies in its applicability to various types of trojan attacks, different label-mapping strategies, and its ability to work with and without clean training data. The authors claim that distorted areas of the learned decision boundary in trojaned models, referred to as blind spots, serve as a consistent signature for distinguishing between trojaned and clean classifiers, regardless of the trojan attack methodology. A key characteristic of these blind spots is that samples within these regions are expected to be out-of-distribution (OOD) relative to the clean training data, yet trojaned classifiers mistakenly identify them as in-distribution (ID) samples. To leverage this characteristic, the authors propose using adversarial attacks to perturb a given OOD sample towards an ID direction, followed by computing the difference in ID score (i.e., maximum softmax probability) before and after the adversarial attack. They found that trojaned models exhibit significantly larger differences in these scores. The effectiveness of the proposed detection method was empirically validated against eight trojan attacks and under different levels of access to clean training data.

**Strengths:**

1. The paper is well-written and easy to read.
2. It includes clear explanations, supported by figures, to illustrate the intuition and proposed method.
3. The authors conduct extensive experiments across various types of trojan attacks and different levels of access to clean training data.
4. They propose a simple yet effective method to detect trojaned models.
5. Theoretical analysis is provided, demonstrating that a trojaned neural network is more sensitive to adversarial perturbations.

**Weaknesses:**

As mentioned in the limitation section, the selections of $\epsilon$ and $\tau$ rely on a validation set and a surrogate model. As a result, the quality of these hyperparameters depend on the choices of the validation set and the surrogate model. In particular, when detecting a trojaned model with a new model architecture or trained on a new domain, one might need to tune $\epsilon$ and $\tau$.

**Questions:**

1. Could you clarify what it means for a trojaned classifier to be adversarially trained on poisoned training data? (For example, for poisoning samples generated by a label-consistent trojan attack, what does it mean by adversarially training the trojaned model on these data?) Additionally, are there any empirical results demonstrating the effectiveness of the proposed method in this specific scenario?
2. When a portion of clean training data is not accessible, the authors propose to use Tiny-ImageNet. In addition, it comes to my mind that given the accessibility to the model, one could actually reverse engineer the training data. In that way, you could obtain a portion of fake clean training data, followed by creating near-OOD samples.Consequently, the performance without clean training data could approach that of having a portion of real clean training data, thereby diminishing the performance gap between TRODO-Zero and TRODO.
In particular, the reverse engineering of training data has been adopted in many backdoor works, such as [1] (see Figure 3).
3. Headings of paragraphs could be set in a uniform format. Eg., In Line 149, there is a ‘.’ after the paragraph heading; by comparison, in Line 168, there is not.

[1] Liu, Y., Ma, S., Aafer, Y., Lee, W. C., Zhai, J., Wang, W., & Zhang, X. (2017). Trojaning attack on neural networks.

**Limitations:**

The performance of the proposed method drops by a large margin (over 10%) when no clean training data is accessible, compared to that with a portion of clean training data.

---

> ### Author Rebuttal · Authors · 2024-08-05
>
> Thank you very much for your positive feedback on our paper. We greatly appreciate your insights and suggestions.
>
> > **W1:**
>
> We acknowledge that extracting hyperparameters may be considered a limitation of our method, as previously discussed in our limitations section. However, it is important to note that these hyperparameters must be selected based on the architecture where the validation is fixed. Moreover, our method demonstrates robustness with respect to the validation set, as evidenced in Table 3 of the Ablation Study. Specifically, using small datasets like FMNIST or SVHN for validation, our method outperforms others by up to 10%.
>
> ---
>
> > **Q1:**
>
> We sincerely apologize for any confusion caused. To clarify, the training dataset is initially compromised using a backdoor attack. The classifier is then adversarially trained using the PGD (Projected Gradient Descent) attack method. As mentioned in the caption of Table 1, all columns labeled ACC* pertain to adversarially trained models. Additionally, lines 287-290 states: "Specifically, TRODO achieves superior performance with an 11.4% improvement in scenarios where trojan classifiers have been trained in a standard (non-adversarial) setting and a 24.8% improvement in scenarios where trojan classifiers have been adversarially trained," indicating a significant improvement in adversarially trained models. Furthermore, Table 2 includes columns for rounds 3, 4, and 11 of TrojAI, which involve adversarially trained models.
>
> Table 1: Scanning performance of TRODO compared with other methods, in terms of Accuracy on
> standard trained evaluation sets (ACC %) and adversarially trained ones (ACC* %).
>
> |**Label Mapping**|**Method**|**MNIST**|**CIFAR10**|**GTSRB**|**CIFAR100**|**Pubfig**|***Avg.***|
> |-|-|-|-|-|-|-|-|
> |||ACC/ACC*|ACC/ACC*|ACC/ACC*|ACC/ACC*|ACC/ACC*|ACC/ACC*|
> |**All-to-One**|||||||||
> ||NC|54.3/49.8|53.2/48.4|62.8/56.3|52.1/42.1|52.5/40.2|55.0/49.4|
> ||ABS|67.5/69.0|64.1/65.6|71.2/65.5|56.4/54.2|56.3/58.3|63.1/62.5|
> ||PT-RED|51.0/48.8|50.4/46.1|58.4/57.5|50.9/45.3|49.1/47.9|52.0/49.1|
> ||TABOR|60.5/45.0|56.3/44.7|69.0/53.8|56.7/45.5|58.6/44.2|60.2/46.6|
> ||K-ARM|68.4/55.1|66.7/54.8|70.1/62.8|59.8/50.9|60.2/47.6|65.0/54.2|
> ||MNTD|57.4/51.3|56.9/52.3|65.2/55.9|54.4/48.8|56.7/50.0|58.1/54.7|
> ||MM-BD|85.2/65.4|77.3/57.8|79.6/65.2|88.5/74.0|65.7/48.3|79.3/62.1|
> ||UMD|81.1/61.2|77.5/54.7|81.4/68.2|69.0/56.3|67.9/49.7|75.4/58.0|
> ||**TRODO-Zero**|80.9/79.3|82.7/78.5|84.8/83.3|75.5/73.7|73.2/70.6|79.4/77.0|
> ||**TRODO**|91.2/89.6|91.0/88.4|96.6/93.2|86.7/82.5|88.1/83.0|***90.7/87.3***|
> |**All-to-All**|||||||||
> ||NC|26.7/21.6|24.9/19.6|31.6/23.2|15.4/11.8|16.8/12.3|23.1/17.7|
> ||ABS|32.5/34.1|30.7/28.8|23.6/20.5|34.3/34.8|31.0/28.2|30.4/29.3|
> ||PT-RED|41.0/33.5|39.6/33.1|45.4/43.9|20.3/15.2|12.6/9.8|31.8/27.1|
> ||TABOR|51.7/39.7|50.2/37.8|48.3/39.5|39.4/30.2|38.6/30.8|45.6/35.6|
> ||K-ARM|56.8/49.7|54.6/47.6|57.5/48.9|51.3/45.0|50.6/47.3|54.2/47.7|
> ||MNTD|27.2/25.2|23.0/18.6|16.9/12.8|29.8/31.0|22.3/17.9|23.8/21.1|
> ||MM-BD|54.3/40.4|49.4/35.1|57.9/44.0|40.7/32.3|41.2/34.1|48.7/37.2|
> ||UMD|82.5/61.9|74.6/60.1|84.2/64.5|70.6/49.9|68.7/52.3|76.1/57.7|
> ||**TRODO-Zero**|82.1/80.8|80.4/77.3|83.8/88.6|74.8/72.3|75.0/75.4|79.2/78.8|
> ||**TRODO**|90.0/87.4|89.3/87.5|92.6/89.1|82.4/85.0|83.2/80.9|***87.5/86.1***|
>
>
> Table 2: Comparison of TRODO and other methods on all released rounds of TrojAI benchmark on
> image classification task. For each method, we reported scanning Accuracy and the average time
> scanning time for a given classifier.
> |**Method**|**Round0 Accuracy/Time(s)**|**Round1 Accuracy/Time(s)**|**Round2 Accuracy/Time(s)**|**Round3 Accuracy/Time(s)**|**Round4 Accuracy/Time(s)**|**Round11 Accuracy/Time(s)**|
> |-|-|-|-|-|-|-|
> |NC|75.1/574.1|72.2/592.6|-/>23000|-/>23000|-/>20000|N/A/N/A|
> |ABS|70.3/481.9|66.8/492.5|62.0/1378.4|70.8/1271.4|76.3/443.2|N/A/N/A|
> |PT-RED|85.0/941.6|84.3/962.7|58.2/>23000|65.7/>25000|66.1/>28000|N/A/N/A|
> |TABOR|82.8/974.2|80.3/992.5|56.2/>29000|60.8/>27000|58.3/>32000|N/A/N/A|
> |K-ARM|91.3/262.1|90.0/283.7|_76.0_/1742.8|**79.0**/1634.1|_82.0_/1581.4|N/A/N/A|
> |MM-BD|68.8/_226.4_|73.2/_231.3_|55.8/_174.3_|52.6/_182.6_|54.1/_178.1_|_51.3_/_1214.2_|
> |UMD|80.4/>34000|79.2/>34000|75.2/>18000|61.3/>19000|56.9/>90000|N/A/N/A|
> |**TRODO**|_86.2_/152.4|_85.7_/194.3|78.1/107.2|_77.2_/122.4|82.8/117.8|**61.3**/**984.3**|
>
> -----
>
> > **Q2:**
>
> We thank the reviewer for the suggestion. We should note that access to a portion of the training data would enable us to create near-OOD samples with higher quality compared to similar methods. Specifically, the mentioned method's strategy for exploiting data from a classifier leads to samples with artifacts and shortcuts, reducing its effectiveness as it becomes more apparent to the model that they do not belong to ID (considering them far OOD). However, to further explore your suggestion, we evaluated both the proposed method and the FastDFKD (Fang Data-free 2022) method for crafting data in situations where there is no access to training data:
>
> |**LabelMapping**|**Method**|**MNIST**|**CIFAR10**|**GTSRB**|**CIFAR100**|**Pubfig**|***Avg.***|
> |-|-|-|-|-|-|-|-|
> ||||ACC/ACC\*|ACC/ACC\*|ACC/ACC\*|ACC/ACC\*|***ACC/ACC\****|
> |All-to-One|TrojaNN|70.2/68.0|72.3/70.1|73.1/71.3|65.5/63.1|63.9/60.8|***69.0/66.7***|
> ||FastDFKD|77.4/75.0|79.2/75.5|80.4/78.0|72.9/70.1|71.2/68.0|***76.2/73.3***|
> ||**TRODO**|91.2/89.6|91.0/88.4|96.6/93.2|86.7/82.5|88.1/83.0|***90.7/87.3***|
>
> ---
>
>
> > **Q3:**
>
> Thank you for pointing out the formatting inconsistency. We apologize for this oversight and will ensure that all paragraph headings are uniformly formatted in the final manuscript.
>
> ---
>
> > **L1:**
>
> We acknowledge that the limitation you mentioned is valid. However, the scenario of having no clean training data is a special case. Most research studies, including ours, typically utilize at least a small amount of clean data to validate their methods.

---

> > ### Comment · Reviewer_T6nS · 2024-08-10
> >
> > W1: Thanks for the clarification.
> >
> > Q1: I still have some confusions about adversarially trained models. "To clarify, the training dataset is initially compromised using a backdoor attack. The classifier is then adversarially trained using the PGD (Projected Gradient Descent) attack method."—Could you clarify: 1) the data used for this adversarial training, 2) along with the objective function, and 3) the motivation for this adversarial training? Thanks.
> >
> > Q2: 1) So according to your claim, can we say that the samples generated by my mentioned's method are less effective than some randomly selected samples from a OOD dataset, e.g. Tiny ImageNet? 2) Is FastDFKD (Fang Data-free 2022) method using models to recover some training data? 2) I think my concern for using models to recover some training data when there is no clean training data arises from the large accuracy gap between TRODO and TRODO-Zero. (My intuition is that the samples generated in this way are more effective than some randomly selected samples from a OOD dataset) Thus, the comparison should be with TRODO-Zero, instead of TRODO.

---

> > > ### Author Response · Authors · 2024-08-11
> > >
> > > Thanks for your feedback. Here are the answers to your questions:
> > > Q1:
> > > 1) Each column in the table 1, above, shows the original training data. The data is first altered by the backdoor attack by adding the backdoor to the input and changing the corresponding output labels based on all-to-one (upper half) or all-to-all (lower half) . Then, adversarial training is applied on the entire dataset including the poisoned and clean samples. Here, each training data, which might be poisoned, undergoes an adversarial attack during the training. As in the regular adversarial training, we only perturb the input and keep the ground truth label unchanged.
> > >
> > > 2) The objective function in the adversarial training is the regular cross-entropy (line 593).
> > >
> > > 3) The main motivation for evaluating scanning methods on such models is that many previously proposed signatures for trojan scanning experience a detection accuracy drop in the case of an adversarially trained model (lines 37-39). For instance, in the table 1 above, UMD's detection acc. drops by almost 20% points on MNIST and CIFAR-10 on the adversarially trained models compared to the standard models (compare ACC with ACC*). The possible reason behind this effect is that adversarial training makes shortcut learning difficult,  potentially leading to harder trigger reverse-engineering and identification. While previous scanning methods struggle with this issue, our proposed method does not rely on such practices and shows insignificant drop under adversarially trained models.
> > >
> > > Q2:
> > > 2, 3) Please note that FastDFKD (Up to 100× Faster Data-free Knowledge Distillation by Fang et al.) is an efficient method of generating synthetic samples given access to a trained model (see Fig. 3 of that paper). Here, we use this model to create surrogates of the training samples and use them in our method TRODO-Zero to enhance it. So indeed the row mentioning "FastDFKD" *is* the TRODO-Zero with the synthesized samples using FastDFKD. As can be seen, there is still a large gap between the two TRODO and TRODO-Zero (it even got slightly worse). The first row, TrojaNN, (Trojan Signatures in DNN Weights, by Fields et al.), which also assumes no access to the training data is included as a baseline.
> > >
> > > 1) Like we mentioned, we hypothesize that the generated samples, through methods such as FastDFKD, could contain shortcuts, potentially imperceptible, making them *unideal* OOD instances, compared to the clean tiny-ImageNet samples that lack such artifacts. Please note that such artifacts would help the classifier to classify the intended synthetic OOD sample as ID more *easily,* reducing the effectiveness of the method, because we take the gap between the ID score of the OOD sample before and after the attack as our detection score (line 179). Therefore, while your idea could generally be beneficial in other methods, it would be less so in our specific method that relies on measuring ID-ness of synthetic OOD samples before and after the attack.

---

> > > > ### Comment · Reviewer_T6nS · 2024-08-11
> > > >
> > > > Q1: Thanks for the detailed clarification. Now I understand the motivation for the adversarial training. I also appreciate the good detection performance of your method compared with others under the adversarial training scenario.
> > > >
> > > > Q2: Thanks for your clarification, especially "the row mentioning "FastDFKD" is the TRODO-Zero with the synthesized samples using FastDFKD.", which solves my confusion. I also appreciate your intuitive explanation in analyzing the bad performance of using the generated training data.
> > > >
> > > > I will keep my score. Thanks!

---

### Official Review · Reviewer_PnTa · 2024-07-14

**Soundness:** 3
**Presentation:** 3
**Contribution:** 3
**Rating:** 3
**Confidence:** 3

**Summary:**

The paper proposes TRODO, a method for identifying trojaned classifiers, which relies on the intuition that in presence of a backdoor it should be easier than for clean classifiers to make the model classify an out-of-distribution input as in-distribution by adding an adversarial perturbation. In practice, a PGD attack is run on OOD images to increase the confidence of the classifier predictions, and the average increase is used to detect backdoored models. In the experimental evaluation, the proposed method outperforms the baselines, being effective against several backdoor attacks even when combined with adversarial training.

**Strengths:**

- The proposed method is very general, as it doesn't make assumptions on the type of backdoor attacks or target architectures. Moreover, the paper proposes variants with and without access to training data. Finally, the proposed approach appears less expensive than the baselines (Table 2).

- The experimental results support the proposed method, which is effective against several backdoor attacks even when combined with adversarial training.

**Weaknesses:**

- The proposed method seems to have brittle elements:
  - the PGD-10 attack used for computing the signature might be considered weak (only 10 iterations), then not able to fully optimize the target loss. Then, a stronger attack could further increase the confidence of the clean classifiers on OOD points, making them more similar to trojaned one (since the confidence is upper bounded, one could get the same score for all classifiers in the worst-case). Thus, the effectiveness of TRODO seems to rely to some extent on the attack not fully optimizing the target loss (which might be even exploited to bypass the detection mechanism).
  - the confidence of any classifier might be adjusted post-training by e.g. temperature rescaling without changing its decisions. In this way, it seems possible to bypass the detection by making a model under-confident.
  - adversarial training variants, e.g. on OOD data [60] or to have uniform predictions far from ID data [A], have been explored in prior works, and might be used to counter the proposed scanning scheme.

- It's not clear how the effectiveness of TRODO correlates with the strength of the backdoor attacks. For example, one can imagine that using a lower poisoning rate might make the attack less detectable (but less effective).

[A] https://arxiv.org/abs/1910.06259

**Questions:**

- Why using a left-truncated normal distribution for estimating $\tau$ when the score S can take values only on a specific range (confidence is upper bounded by 1)?

- While the proposed method provides good experimental results, I think it's important to address the concerns about its robustness (see above).

**Limitations:**

Limitations are partially addressed.

---

> ### Author Rebuttal · Authors · 2024-08-04
>
> Thank you for your thoughtful review. We have provided the following response:
>
> >**W1.1:**
>
> We believe the epsilon value of the attack plays a key role compare to steps in our setup. If epsilon is large, as you mentioned, our signature for both clean and Trojan classifiers would be the same. This is the main reason we carefully estimate it using the validation set and Boundary Confidence Level to avoid such scenarios (see line 199). The number of steps in the attack has less effect as the attack radius has already been determined (Madry et al. Toward 2017). To further address your concern, we have provided an additional experiment where every component of our pipeline remains fixed except for the number of steps of the attack:
>
>
>
> |**Label Mapping**|**Method**|**n-step**|**MNIST**|**CIFAR10**|**GTSRB**|**CIFAR100**|**Pubfig**|***Avg.***|
> |-|-|-|-|-|-|-|-|-|
> ||||ACC/ACC*|ACC/ACC*|ACC/ACC*|ACC/ACC*|ACC/ACC*|***ACC/ACC****|
> |**All-to-One**|||||||||
> ||TRODO|PGD-10 **(default)**|91.2/89.6|91.0/88.4|96.6/93.2|86.7/82.5|88.1/83.0|***90.7/87.3***|
> |||PGD-100|91.0/89.0|90.2/87.6|95.9/92.7|86.0/81.9|87.6/82.1|***89.9/86.6***|
> |||PGD-1000|90.5/89.1|90.5/87.6|96.3/92.7|86.1/81.7|87.9/82.4|***90.5/86.4***|
>
> ---
>
>
> > **W1.2:**
>
>
> It is possible that temperature rescaling affects the regular softmax function, but the classifier remains more confident in in-distribution samples compared to OOD samples (Tajwar et al., No True 2021). This is consistent with the rationale and principle behind our method. As a result TRODO, would also detect trojaned classifiers with temperature scaling.  To further clarify this concern, we kindly ask you to refer to our _common response_, where we considered worst-case scenarios involving adaptive attackers. Even when the attacker is aware of our defense mechanism, TOROD demonstrated robust and consistent performance.
>
>
> ---
> > **W1.3:**
>
> The mentioned papers, CCAT (Stutz et al., 2020) [A] and RATIO (Augustin et al., 2020) [60], aimed to enhance robust performance on in-distribution samples. Improving robustness on adversarially out-of-distribution (OOD) samples was not their main purpose. Although they considered the issue of overconfidence on OOD samples, their primary goal was robust in-distribution classification. They do not weaken principle behind TRODO, which is based on the vulnerability of existing robust methods, including CCAT, RATIO, and other SOTA methods, to perturbed OOD samples, especially when they are close to the in-distribution, referred to as near OOD in our paper (see line 88 and Table 5). This observation has also been demonstrated by Fort (Adversarial, **2022**). Moreover, recently, RODEO (ICML, **2024**) has shown that in the CIFAR10 vs. CIFAR100 OOD detection challenge, there is no better performance than random detection (i.e. less than 50% AUROC), which further supports our claim (see their Table 4-a of our paper). Additionally, we kindly ask you to check our _common response_, where we show that even in the worst-case scenario (adaptive attack), TRODO demonstrates promising performance.
>
> _Finally, to further address your concern, we trained input trojan models using the method proposed by RATIO instead of common adversarial training and reported TRODO's performance on them here. Other components of TRODO, including the training pipeline, remain fixed._
>
>
>
> |Label Mapping|Method|MNIST|CIFAR10|GTSRB|CIFAR100|Pubfig|***Avg.***|
> |-|-|-|-|-|-|-|-|
> |||ACC/ACC*/ACC**|ACC/ACC*/ACC**|ACC/ACC*/ACC**|ACC/ACC*/ACC**|ACC/ACC*/ACC**|***ACC/ACC*/ACC\*\****|
> |All-to-One|||||||||
> ||TRODO-Zero|80.9/79.3/78.4|82.7/78.5/79.1|84.8/83.3/82.8|75.5/73.7/74.0|73.2/70.6/70.2|***79.4/77.1/76.9***|
> ||TRODO|91.2/89.6/89.4|91.0/88.4/88.7|96.6/93.2/92.4|86.7/82.5/81.5|88.1/83.0/83.4|***90.7/87.3/87.1***|
>
>
>
>
>
> ---
>
> >**W2:**
>
> Intuitively, increasing the poisoning rate enlarges the blind spots in trojaned classifiers, as these are boundary regions where the poisoned data causes the model to overfit. Consequently, this will increase the probability that TRODO detects the trojaned classifiers. However, our signature is based on the presence of blind spots in trojaned classifiers and shows consistent performance across different poisoning rates. In the paper, we considered a poisoning rate of 10% as it is common in the literature. In response to your comment, we have provided TRODO's performance for different rates below: _(Other components of TRODO remained fixed.)_
>
> |**Label Mapping**|**Method**|**Poisoning-Rate**|**MNIST**|**CIFAR10**|**GTSRB**|**CIFAR100**|**Pubfig**|***Avg.***|
> |-|-|-|-|-|-|-|-|-|
> ||||ACC/ACC*|ACC/ACC*|ACC/ACC*|ACC/ACC*|ACC/ACC*|***ACC/ACC\****|
> |**All-to-One**|||||||||
> ||TRODO-Zero|1%|80.1/78.2|81.3/77.0|83.5/81.8|74.6/72.7|72.1/69.8|***78.3/75.9***|
> |||3%|82.0/79.3|82.5/78.1|85.4/83.2|76.7/74.4|74.0/71.1|***80.1/77.2***|
> |||5%|83.6/80.4|84.0/79.5|86.3/84.6|77.5/75.3|75.1/72.6|***81.3/78.5***|
> |||10% **(default)**|80.9/79.3|82.7/78.5|84.8/83.3|75.5/73.7|73.2/70.6|***79.4/77.0***|
> ||TRODO|1%|89.5/87.4|88.7/85.6|94.9/91.0|84.5/81.2|86.4/80.3|***88.8/85.1***|
> |||3%|91.0/89.2|90.5/87.8|96.5/93.1|86.3/82.7|87.8/82.4|***90.4/87.0***|
> |||5%|92.8/90.6|91.7/88.9|97.0/94.1|87.5/83.6|89.1/84.5|***91.6/88.3***|
> |||10% **(default)**|91.2/89.6|91.0/88.4|96.6/93.2|86.7/82.5|88.1/83.0|***90.7/87.3***|
>
>
> ----
>
> > **Q1:**
>
> Thank you for your insightful question. You are correct that the score $S_i$ can take values only within a specific range, as confidence is upper bounded by $1$. We apologize for the oversight in the paper where we did not mention that we apply a transformation to the score $S_i$.
> In our method, we use $-log(1 - S_i)$ instead of $S_i$ directly. This transformation maps the original score to a new range $[0, ∞)$, making it suitable for fitting with a left-truncated normal distribution.
> We will include this clarification in the final manuscript to ensure there is no confusion.

---

> > ### Comment · Reviewer_PnTa · 2024-08-11
> >
> > Thanks for the detailed response and additional experiments.
> >
> > > **W1.2:** "the classifier remains more confident in in-distribution samples compared to OOD samples"
> >
> > If I understand it correctly, only OOD samples are used to compute the signature. Then, I think the absolute difference $\textrm{ID-Score}_f (x_i^{OOD*}) − \textrm{ID-Score}_f (x_i^{OOD})$ could be (most likely) made arbitrarily small with temperature rescaling, so that it's in the range of the signature of a clean classifier. Similarly, since the confidence is upper bound by 1, also an over-confident classifier which has near 1 confidence on OOD data would lead to very small differences between adversarially perturbed and clean OOD points (since the ID-Score cannot increase much). Am I missing something?
> >
> > > **W1.3:** "Improving robustness on adversarially out-of-distribution (OOD) samples was not their main purpose"
> >
> > RATIO loss includes a term which is a robust loss on OOD data, then it directly optimizes adversarial robustness on OOD points. What was the OOD data used for training the RATIO models? In general, these methods show that it is possible to control the (adversarial) confidence on OOD data (but might require some adaptation to the specific task of bypassing the detection mechanism).
> >
> > > common response
> >
> > I think the additional experiments with the new losses show that adaptive attacks have the potential to bypass the detection mechanism of TRODO: in fact, $L_{adaptive1}$ improves 12% compared to $L_{default}$ in the All-to-All setup, which seems significant.

---

> ### Author Response · Authors · 2024-08-11
> **Response to Reviewer PnTa**
>
> > **Am I missing something?**
>
>
>
> * We believe there has been some misunderstanding. When we mentioned that _'the classifier remains more confident in in-distribution samples compared to OOD samples,'_ we were referring to **Fact-A**. We would like to clarify that the principle behind our signature remains valid even when temperature rescaling is applied. The logical flow of our principle is as follows: **Fact-A ⇒ Fact-B ⇒ Fact-C ⇒ Fact-D ⇒ Fact-E ⇒ Fact-F ⇒ Fact-G**.
>
> * Furthermore, as indicated in Fact-A, the softmax output tends to resemble a uniform distribution for OOD samples. As a result, applying temperature rescaling will affect all logits equally and **will not**  alter this uniform shape. Consequently the ID score for clean OOD samples would still be very low (e.g., approximately $\frac{1}{N}$, where $N$ is the number of ID classes). However, by perturbing these samples, the ID score can increase (up to one). Therefore, our signature would not be arbitrarily small as mentioned; it remains within the range of ([0,$1-\frac{1}{N}$]).
>
>
> **Fact-A**: A classifier remains more confident in IDs compared to OODs. Specifically, the output softmax of a classifier for OODs tends to be more uniform, while for IDs, it is more concentrated.  (confidence: maximum softmax probablity) [1].
>
>
> **Fact-B**: Previous research has primarily focused on perturbing IDs, shifting them, and using this as a signature to distinguish between clean and trojaned classifiers [2,3,4,5,6,7,8,9,10].
>
>
> **Fact-C**: In scenarios where classifiers have been adversarially trained, perturbing ID samples for signature extraction becomes less effective. This is because adversarial training enhances the classifier's robustness to such perturbations, reducing their impact [9,10].
>
>
> **Fact-D**: Instead of perturbing IDs, perturbing OODs toward the in-distribution is a viable approach, as classifiers are vulnerable to this shift. This approach remains effective even for adversarially trained classifiers, as they are still susceptible to perturbed OODs. This finding is also supported by parallel research in OOD detection [11,12,13,14].
>
>
> **Fact-E**: Trojaned classifiers learn decision boundaries that include 'blind spots,' which are intuitively regions along the in-distribution boundary that have been altered to overfit on poisoned training data (data with triggers), thereby changing the geometry of the boundary.
>
>
> **Fact-F**: Perturbing OODs toward the classifier's decision boundary increases their ID scores, and this effect is particularly significant in trojaned classifiers. This is because the perturbations can exploit the blind spots within the decision boundary, mimicking the triggers used during the trojan attack.
>
> **Fact-G***: We use the difference in ID scores between a benign OOD sample (without perturbation) and an adversarially perturbed OOD sample as a signature to differentiate between clean and trojaned classifiers.
>
>
>
>
> ---
>
> >**What was the OOD...?**
>
> Following their reported setting, we used Tiny ImageNet as the out-of-distribution dataset for training.
>
> ---
>
> >**In general, these methods.. .**
>
>
> Regardless of the adaptation strategy, existing classifiers are vulnerable to adversarial attacks on OODs, particularly when these samples are close to the in-distribution boundary. This vulnerability, as demonstrated recently by (Lorenz Deciphering, 2024), (Fort Adversarial Vulnerability, 2022), and others [11,12,13,14], limits them to act as counters to our method. To further illustrate this, we conducted additional experiments using RATIO as an OOD detector, where the MSP is employed as the ID score. In the first scenario, CIFAR-10 serves as the ID dataset and CIFAR-100 as the OOD, and vice versa in the second. While the method achieves good performance in clean scenarios , in adversarial scenarios—where perturbations are added to shift OODs to in-distribution and vice versa—its performance drops to below random detection, highlighting its vulnerability to perturbed OODs.
>
> | Method\Benchmark | _CIFAR-10 vs.CIFAR-100_         |_CIFAR-100 vs. CIFAR-10_|
> |-|-|-|
> |              | AUC/AUC*      |AUC/AUC*  |
> | RATIO  | 81.3/14.8     |68.5/9.0  |
>
>
> ---
>
> >**I think the additional ...**
>
>
> In the adaptive attack scenario, we evaluated various label mappings and adaptive strategies. The worst performance decrease we observed was 12% (from 86.0% to 74.1%). Despite this, our method still outperforms previous detection methods that were not subjected to adaptive attacks. For instance, UMD achieves 57.7%, while our approach achieves 74.1%.
>
>
>
> [1] Hendrycks, A baseline, 2016
>
> [2] Xiang, UMD: Unsupervised  , 2023
>
> [3] Wang, Mm-bd: , 2024
>
> [4] Wang, Neural, 2019
>
> [5] Liu, Abs:Scanning, 2019
>
> [6] Shen, Backdoor scanning for, 2021
>
> [7] Guo, Tabor, 2019
>
> [8] Hu, Trigger hunting, 2022
>
> [9] Edraki, Odyssey, 2021
>
> [10]Zhang, Cassandra, 2021
>
> [11] Chen Robust OOD 2020
>
> [12] Azizmalayeri, Your Detector 2022
>
> [13] Chen, Atom ,2021
>
> [14] Mirzaei, RODEO, 2024

---

> > ### Comment · Reviewer_PnTa · 2024-08-12
> >
> > > Furthermore, as indicated in Fact-A, the softmax output tends to resemble a uniform distribution for OOD samples. As a result, applying temperature rescaling will affect all logits equally and will not alter this uniform shape.
> >
> > Unless the softmax output is exactly uniform (i.e. all logits are identical), which seems very unlikely, temperature will change the distribution.
> >
> > > However, by perturbing these samples, the ID score can increase (up to one). Therefore, our signature would not be arbitrarily small as mentioned; it remains within the range of ($[0, 1 - 1/N]$).
> >
> > I meant arbitrarily small within its range of course (as the ID-Score after the attack cannot be lower than on the clean input).
> >
> > The point of using temperature >> 1, i.e. making the model under-confident, is that even when the attack is applied it will not increase much the confidence, in particularly since the hyperparameters of the attack such as $\epsilon$ and number of steps is fixed and calibrated on standard models.
> >
> > Also, one can consider the other extreme case with temperature = 0 (in practice, values close to 0 are sufficient), i.e. the softmax becomes the argmax function. If the argmax of the logits is unique, then the softmax output is a one-hot vector, and even for OOD points the difference of ID-Scores will be 0. This seems to bypass the detection mechanism. Is this correct?

---

> ### Author Response · Authors · 2024-08-12
> **Response to Reviewer PnTa**
>
> We thank the reviewer for their thoughtful review and valuable feedback.
>
> * A key requirement for deploying deep neural networks in real-world applications is calibrated or approximately calibrated behavior on input samples. A miscalibrated classifier, particularly in cases where $T→0$,  $T>>1$, can lead to poor decision-making. For example, in medical diagnosis, overestimating disease probability can result in unnecessary treatments, while underestimating it may lead to missed diagnoses. Miscalibrated models are especially unreliable in high-stakes scenarios such as finance, healthcare, or autonomous systems, where accurate probability estimates are crucial. Furthermore, the backdoor attack/defense literature generally assumes that Trojaned classifiers behave normally on samples without triggers, similarly to standard classifiers. Since standard classifiers typically exhibit approximately calibrated outputs, we implicitly assume that the Trojaned classifier would also retain this characteristic.
>
> * We acknowledge the challenges posed by the extreme temperature scenarios, specifically when $T→0$ or  $T>>1$, as our ID-score would converge to zero in both cases. However, we believe this issue can be addressed with minimal modifications. In the backdoor attack setup (as detailed in our threat model), it is commonly assumed that the attacker has access to the input model. To mitigate the issue, we propose a minor extension to TRODO: applying its own softmax function to the logits of the input classifier instead of relying on the input classifier's final softmax output. This approach could help counter the adversarial temperature settings highlighted by the reviewer.
>
> * Moreover, since TRODO and TRODO-Zero both utilize an OOD  set, another minor extension could involve evaluating the classifier's softmax output on these samples. If the output significantly deviates from a uniform distribution (e.g. measured by Kullback–Leibler divergence), the model could be rejected as extremely miscalibrated due to the attacker's manipulation.
>
>
>
> To further address your concern, we evaluated TRODO's performance on an all-to-one label mapping task using the CIFAR-10 and GTSRB datasets across different temperature values, while keeping other components unchanged. The results show that our designed detector effectively handles a reasonable range of temperature values, maintaining consistent performance throughout.
>
>
> | CIFAR10 | $T=0.5$     |  $T=0.7$    | $T=1$(default)      | $T=1.2 $   |$T=1.5$         |  $T=2$     |
> |-|-|-|-|-|-|-|
> | ACC/ACC*            | 90.5/87.1   | 91.2/87.3   | 91.0/88.4    | 89.4/86.2   | 88.4/85.7    | 87.1/85.3   |
>
>
> | GTSRB | $T=0.5$     |  $T=0.7$    | $T=1$(default)      | $T=1.2 $   |$T=1.5$         |  $T=2$     |
> |-|-|-|-|-|-|-|
> | ACC/ACC*            | 95.2/92.7   | 94.8/93.0 |96.6/93.2   | 95.7/92.1  |94.3/92.8    |92.7/91.5   |

---

> > ### Comment · Reviewer_PnTa · 2024-08-13
> >
> > Thanks for the additional reply.
> >
> > > A miscalibrated classifier [...] can lead to poor decision-making...
> >
> > In classification tasks as those reported in the paper, calibration is not a factor, since confidence is not used for classification (only argmax).
> >
> > > Since standard classifiers typically exhibit approximately calibrated outputs...
> >
> > I think this is not in general precise, that is it's not clear whether classifiers are calibrated before applying post-processing calibration techniques (see e.g. https://arxiv.org/abs/1706.04599).
> >
> > > However, we believe this issue can be addressed with minimal modifications...
> >
> > This might be true, but should be experimentally tested and discussed in the paper.
> >
> > &nbsp;
> >
> > I think this simple approach (modifying the confidence of the model via temperature rescaling), which doesn't require designing new backdoor attacks, points to weaknesses of the proposed detection method, which can potentially be exploited by more sophisticated adaptive attacks. I think this can't be simply dismissed assuming models are well calibrated, and should be discussed in the paper.
> >
> > Overall, the rebuttal has confirmed that the proposed method, in the current form, is susceptible to temperature rescaling and (at least partially) other adaptive attacks ($L_{adaptive1}$). Thus, I think the paper would require significant improvements, at least discussing the current limitations and potential countermeasures. Then, since the original main concerns remain, I will vote for rejecting the paper.

---

> > > ### Author Response · Authors · 2024-08-13
> > > **Response to Reviewer PnTa**
> > >
> > > We thank the reviewer for their feedback.
> > >
> > > We believe the discussion around temperature scaling has diverged from our primary focus. Our intention was to highlight the extreme cases of a miscalibrated classifier, such as those producing one-hot or uniform outputs, which, as the reviewer suggested, are uncommon in real-world scenarios due to their lack of explainability. This is why we did not address them in our paper. While these scenarios may be relevant in theoretical contexts, they are not directly applicable to practical situations. Nonetheless, our adaptive attacks demonstrate that even in the worst-case scenarios, TRODO performs consistently. To **fully address** the reviewer's concerns, as they **agreed**, we will consider applying the softmax function ourselves rather than relying on the classifier's softmax, and we will discuss this in our paper.
> > >
> > > >**This might be true, but should be experimentally tested and discussed in the paper.**
> > >
> > > Our experiments were conducted under the assumption that temperature scaling was not applied ($T=1$). Therefore, applying softmax ourselves instead of using the classifier’s built-in softmax would not change the results, and the experiments would yield identical outcomes.
> > >
> > > >**Overall, the rebuttal has confirmed that the proposed method is susceptible to.... adaptive attacks**
> > >
> > > We believe that a fair comparison should also consider the performance of other methods under adaptive attacks for a evaluation. However, even under strong adaptive attacks, our method continues to demonstrate superior performance compared to existing scanning methods.
> > >
> > >
> > >
> > >
> > > We remain open to further discussion to address any additional concerns the reviewer may have.
> > >
> > > Sincerely,
> > > The Authors

---

### Official Review · Reviewer_eToF · 2024-07-15

**Soundness:** 3
**Presentation:** 2
**Contribution:** 3
**Rating:** 6
**Confidence:** 4

**Summary:**

This paper points out a limitation of existing backdoor model scanning methods:
They fail to detect backdoored models trained with adversarial training. It
propose a new backdoor model scanning method by utilizing adversarial shifts in
Out-of-distribution samples. Experiments on MNIST, CIFAR-10, GTSRB, CIFAR-100,
and Pubfig show the effectiveness of the proposed method.

**Strengths:**

* The investigated problem is interesting.

* The motivation of this paper is good.

* This paper provides theoretical analysis to support the proposed observation
and method.

* The proposed scanning method is general to different types of backdoor attacks.

**Weaknesses:**

* The paper states that experiments were conducted using ResNet18, PreActResNet18,
and ViT-B/16 models. However, Table 1 only presents results for ResNet18. While
the paper claims that results for other models are included in Appendix Section
M, but this section also only contains ResNet18 results. It would be
beneficial to explicitly present the detection accuracy for each model across
various attack scenarios.

* The discussion about the adaptive attacks to the proposed method is missing,
where the attacker knows about the proposed defense strategy and actively tries
to evade or overcome it. For example, the adaptive attacker might able to add a
loss during the backdoored model construction phase to reduce the change of the
ID-Score on the backdoored models.

* While theoretical analysis of the proposed method is highly recommended, the
paper would benefit from more intuitive explanations of this analysis. There
appears to be a lack of detailed clarification regarding the connection between
the theoretical analysis and the proposed method. For instance, the relationship
between Theorem 2 and the fundamental principles of the proposed method is not
clearly stated. Providing more intuitive explanations of these theoretical
analysis would strengthen the paper.

**Questions:**

See Weaknesses.

**Limitations:**

The limitations have been discussed in this paper.

---

> ### Author Rebuttal · Authors · 2024-08-05
>
> We appreciate your insightful review. Here is our detailed response:
>
> >**W1:**
>
> We sincerely apologize for this oversight. There were some issues with the command that removed these tables from our submitted paper. Here, we have provided those tables and assure you that they will be included in the final manuscript. We should note that in the following experiments, all components of TRODO are fixed except for the architectures of the classifiers.
>
> _Below are the detection results in terms of accuracy on standard trained evaluation sets (ACC %) and adversarially trained ones (ACC* %). Due to character limits, we have included only the top competitors from the methods considered in Table 1 of our paper._
>
> **PreAct ResNet-18 Architecture:**
>
> |**LabelMapping**|**Method**|**MNIST**|**CIFAR10**|**GTSRB**|**CIFAR100**|**Pubfig**|***Avg.***|
> |-|-|-|-|-|-|-|-|
> |||ACC/ACC\*|ACC/ACC\*|ACC/ACC\*|ACC/ACC\*|ACC/ACC\*|***ACC/ACC\****|
> |***All-to-One***|||||||||
> ||K-ARM|69.9/54.3|68.2/52.0|72.0/62.8|58.3/48.6|61.3/48.6|***65.9/53.2***|
> ||MM-BD|81.5/67.3|75.6/57.1|75.8/61.1|86.1/72.7|62.4/47.6|***76.3/61.2***|
> ||UMD|79.5/58.8|76.4/51.5|79.2/64.7|67.5/54.6|64.5/47.0|***73.4/55.3***|
> ||**TRODO-Zero**|85.0/78.5|81.2/79.1|85.2/83.9|78.2/78.9|73.6/72.4|***80.6/78.6***|
> ||**TRODO**|92.6/88.7|90.5/90.2|93.4/90.1|85.6/83.2|80.2/78.8|***88.5/86.2***|
> |***All-to-All***|||||||||
> ||K-ARM|56.8/49.7|54.6/47.6|57.5/48.9|51.3/45.0|50.6/47.3|***54.2/47.7***|
> ||MM-BD|52.7/42.3|49.3/35.1|57.0/44.2|41.3/32.0|40.0/34.0|***48.1/37.5***|
> ||UMD|80.7/61.5|75.5/55.9|83.5/64.9|67.7/50.0|66.7/47.5|***74.8/56.0***|
> ||**TRODO-Zero**|83.9/76.8|77.2/78.4|87.5/83.7|78.4/76.8|76.0/70.3|***80.6/77.2***|
> ||**TRODO**|90.8/87.9|88.3/89.8|92.0/87.9|82.6/82.1|81.0/76.6|***86.9/84.9***|
>
> **ViT-B-16 Architecture:**
>
> |**LabelMapping**|**Method**|**MNIST**|**CIFAR10**|**GTSRB**|**CIFAR100**|**Pubfig**|***Avg.***|
> |-|-|-|-|-|-|-|-|
> |||ACC/ACC\*|ACC/ACC\*|ACC/ACC\*|ACC/ACC\*|ACC/ACC\*|***ACC/ACC\****|
> |***All-to-One***|||||||||
> ||K-ARM|69.8/54.3|68.2/52.0|72.0/62.8|58.3/48.6|61.3/48.6|***65.9/53.2***|
> ||MM-BD|72.9/58.4|67.6/49.8|67.5/52.5|78.0/62.7|54.4/39.5|***68.1/52.6***|
> ||UMD|75.2/55.0|69.0/45.4|71.5/59.6|62.5/48.6|58.3/40.7|***67.3/49.8***|
> ||**TRODO-Zero**|78.5/71.8|72.9/71.5|76.5/76.1|71.6/70.2|68.9/65.1|***73.7/71.0***|
> ||**TRODO**|87.9/85.6|82.5/84.4|85.2/84.3|80.3/79.6|78.2/76.1|***82.8/82.0***|
> |***All-to-All***|||||||||
> ||K-ARM|54.9/43.3|50.5/43.1|51.5/47.6|46.4/39.9|49.4/41.9|***50.5/43.2***|
> ||MM-BD|51.9/40.7|44.3/33.1|57.8/41.6|41.0/29.6|40.4/28.0|***47.1/34.6***|
> ||UMD|73.9/56.4|69.4/48.6|77.7/58.6|58.8/43.4|58.0/40.1|***67.5/49.4***|
> ||**TRODO-Zero**|80.2/76.0|70.4/73.3|81.6/77.0|74.2/69.8|71.7/65.8|***75.6/72.4***|
> ||**TRODO**|87.6/82.3|82.6/84.3|83.5/83.0|79.8/77.3|76.0/73.1|***81.9/80.0***|
>
> -----
> > **W2:**
> Thank you for suggesting this evaluation, and we apologize for missing it in our submitted version. To address the mentioned concern, we have provided different scenarios of adaptive attacks in our _common response_, which we kindly ask you to review.
> ---
> > **W3:**
>
> We acknowledge that the paper lacks an intuitive explanation connecting the theoretical analysis to the proposed method.
> The theory section aims to provide high-level intuition for the method's core principle: _trojaned models are more susceptible to adversarial perturbations, especially in near-OOD regions_.
>
> As deducible from our experiments, there is a clear performance gap between TRODO and TRODO-Zero, and the main reason for this gap is the utilization of near-OOD samples instead of arbitrary (mostly far-OOD) ones. We have also illustrated this phenomenon on Figure 2 of the paper. Using near-OOD samples enhances the change of ID-Score, resulting in more recognizable signature for trojan scanning. To further validate this intuition, we have provided Theorem 1, in which we prove that the adversarial risk on near-OOD data is more evident.
>
> In Theorem 2, we analyzed a simplified scenario using a least-square loss and two-layer networks. Despite the simplicity, these cases are indicative of the general scenario because other complex losses used in practice yield similar optimization outcomes. Additionally, two-layer networks are known to be universal approximators capable of learning any function, analogous to more complex architectures like ResNet with MSE loss. This theoretical foundation helps to explain the observed phenomena in more intricate setups, thereby reinforcing the validity of our proposed method. Regarding the connection of this Theorem with our work, it is noteworthy that the core principle of TRODO is to use the difference of ID-Score of OOD samples before and after attack as the scanning signature. This change of ID-Score is equivalent to the adversarial risk which we have defined in Section 4. According to this theorem, this risk is linearly bounded by the trigger norm ($||t||$), which is non-zero for backdoored models and 0 for clean ones, making them distinguishable by our signature.
>
> In future revisions, we will elaborate on this connection to provide readers with a clearer understanding of how our theoretical analysis supports and motivates the proposed method.

---

> > ### Comment · Reviewer_eToF · 2024-08-12
> >
> > Thanks for your detailed rebuttal. As most of my concerns are addressed, I will increase my rating to 6.

---

> > > ### Author Response · Authors · 2024-08-12
> > > **Appreciation for Your Feedback and Review**
> > >
> > > Thank you for your valuable review and positive feedback! We are pleased to hear that your concerns have been addressed.
> > >
> > > Warm regards,
> > > The Authors

---

### Official Review · Reviewer_3HiF · 2024-07-18

**Soundness:** 3
**Presentation:** 3
**Contribution:** 2
**Rating:** 5
**Confidence:** 4

**Summary:**

The paper introduces a novel trojan scanning method named TRODO (TROjan scanning by Detection of adversarial shifts in Out-of-distribution samples). TRODO leverages the concept of "blind spots," where trojaned classifiers mistakenly identify out-of-distribution (OOD) samples as in-distribution (ID). The method scans for these blind spots by adversarially shifting OOD samples towards in-distribution, using the increased likelihood of these perturbed samples being classified as ID as a signature for trojan detection. TRODO is both trojan and label mapping agnostic, effective even against adversarially trained trojaned classifiers, and applicable even when training data is absent.

**Strengths:**

The writing is clear and the pictures are crisp and clear.

**Weaknesses:**

Undefined Threat Model: The threat model is not clearly defined, and the adversary's capabilities are not explicitly listed.

Lack of Comparison with SOTA Baselines: There is an absence of comparison with SOTA baselines, such as FreeEagle. The results listed in Table 2 (baseline-NC) differ significantly from those in FreeEagle's Table 4. Additionally, many baselines from FreeEagle's Table 4, such as STRIP[2] and ANP[3], are not discussed.

[1]Fu, C., Zhang, X., Ji, S., Wang, T., Lin, P., Feng, Y., & Yin, J. (2023). {FreeEagle}: Detecting Complex Neural Trojans in {Data-Free} Cases. In 32nd USENIX Security Symposium (USENIX Security 23) (pp. 6399-6416).
[2]Gao, Y., Xu, C., Wang, D., Chen, S., Ranasinghe, D. C., & Nepal, S. (2019, December). Strip: A defence against trojan attacks on deep neural networks. In Proceedings of the 35th annual computer security applications conference (pp. 113-125).
[3]Wu, D., & Wang, Y. (2021). Adversarial neuron pruning purifies backdoored deep models. Advances in Neural Information Processing Systems, 34, 16913-16925.

**Questions:**

1. Can you clarify your threat model?
2. Why do you have so many missing baselines?

**Limitations:**

Design is too simple

---

> ### Author Rebuttal · Authors · 2024-08-05
>
> Thank you for your valuable comments. Our responses to each point are provided below:
>
>
>
> >**Q1&Undefined Threat Model**
>
> Sorry for the confusion. We have briefly stated our threat model in lines 212-226 of the paper. We further present our threat model more clear in depth here. We assure the reviewer that we will improve the clarity of this section in our final manuscript.
>
> ### **Attacker Capabilities**
> - **Data Poisoning and Training Influence:** Attackers can poison training data [1, 3] and influence the training process [2, 4] to embed backdoors into models.
> - **Trigger Visibility and Coverage:**
>   - **Stealthy to Overt Modifications:** Attackers can deploy triggers that range from undetectable to more noticeable modifications.
>   - **Local and Global Coverage:** Triggers can affect specific parts of a sample [1, 4] or the entire sample [5, 6].
>   - **Sample-Specific Attacks:** Attacks can be tailored to specific samples [7], complicating detection.
> - **Label-Consistent Mechanisms:** Attackers can use poisoned inputs labeled according to their visible content, which leads to misclassification during inference [8, 5].
> - **Attack Types:** Attacks can be either All-to-One or All-to-All. In the former, a single target class is selected and whenever the input contains the trigger, the classifier outputs the selected target class. In the latter however, there is more control over the attack. For each source class (the class to which the clean input actually belongs), an arbitrary target class can be chosen so that the presence of the trigger causes the model to classify the input as that target class.
> - **Model Training:** Models can be trained adversarially or non-adversarially.
>
> ### **Attacker Goals**
> - **Embed Backdoors:** Ensure the model contains backdoors that lead to misclassification during inference.
> - **Maintain Stealthiness:** Create triggers that may be undetectable or hard to detect, even under scrutiny.
> - **Evade Detection:** Implement attacks that complicate detection efforts, especially those that are sample-specific or label-consistent.
>
> ### **Defender Capabilities**
> - **Model-Only Detection:** The defender receives the model and may (TRODO) or may not (TRODO-Zero) have access to a small set of clean samples from the same distribution as the training data.
> - **No Prior Knowledge Required:** The detection mechanism operates without any prior knowledge of the specific type of attack or the nature of the trigger involved.
>
> ### **Defender Goals**
> - **Detect Backdoors:** Identify if the model has been compromised with a backdoor.
> - **Adapt to Various Scenarios:** Effectively scan and detect backdoors both in scenarios with and without clean training samples.
>
> ---
>
> >**Q2&Lack of Comparison with SOTA Baselines**
>
> We have compared TRODO with eight strong Trojan detector methods in our paper. To further address your concerns regarding the comparison with other baselines, we have conducted additional experiments, including comparisons with FreeEagle, STRIP, ANP, Ex-Ray, and DF-TND. The results of these comparisons are presented in the following table:
>
> |**LabelMapping**|**Method**|**MNIST**|**CIFAR10**|**GTSRB**|**CIFAR100**|**Pubfig**|***Avg.***|
> |-|-|-|-|-|-|-|-|
> |||ACC/ACC\*|ACC/ACC\*|ACC/ACC\*|ACC/ACC\*|ACC/ACC\*|***ACC/ACC\****|
> |***All-to-One***|||||||||
> ||DF-TND|56.6/54.8|49.1/49.7|60.2/57.5|49.6/45.5|47.8/42.2|***52.7/49.9***|
> ||STRIP|63.3/71.0|68.4/64.4|71.2/61.2|59.4/53.5|57.2/55.6|***63.9/61.1***|
> ||ANP|78.5/59.7|66.5/51.9|73.4/65.8|76.2/69.3|61.8/47.7|***71.3/58.9***|
> ||Ex-Ray+ABS|66.6/46.5|63.3/51.4|76.9/56.5|60.5/46.4|60.5/50.2|***65.6/50.2***|
> ||FreeEagle|80.2/72.9|82.0/73.2|81.0/82.3|73.2/66.9|65.0/66.0|***76.3/72.3***|
> ||**TRODO-Zero**|80.9/79.3|82.7/78.5|84.8/83.3|75.5/73.7|73.2/70.6|***79.4/77.0***|
> |***All-to-All***|||||||||
> ||DF-TND|23.8/20.7|28.9/26.7|30.9/26.6|13.8/10.1|15.2/12.5|***22.5/19.3***|
> ||STRIP|33.1/28.2|26.5/24.3|22.8/21.1|33.8/29.9|29.4/27.6|***29.1/26.2***|
> ||ANP|52.4/47.9|44.5/40.8|52.7/48.4|42.5/38.2|36.1/32.5|***45.6/41.6***|
> ||Ex-Ray+ABS|47.4/45.4|48.3/44.7|52.6/50.9|39.3/35.7|38.2/33.3|***45.2/42.0***|
> ||FreeEagle|79.8/75.2|54.9/50.2|55.2/52.9|56.5/52.7|48.0/46.1|***58.9/55.4***|
> ||**TRODO-Zero**|82.1/80.8|80.4/77.3|83.8/88.6|74.8/72.3|75.0/75.4|***79.2/78.8***|
>
> Although FreeEagle aims to be effective against various types of trojan attacks and performs better than other baselines in this table, it is particularly vulnerable in All-to-All settings, where samples from each class can be mapped to different target classes in the presence of a trigger. FreeEagle primarily addresses the scenario where a single source class is mapped to a single target class and attempts to identify such pairs, if they exist. Thus, it can only perform well in All-to-One scenarios.
>
> ---
> >**The results listed in Table 2 (baseline-NC) differ significantly ...**
>
> Regarding the results of NC on the TrojAI benchmark (Table 2 in our paper), it is important to note that our results are specific to the TrojAI benchmark, which is not covered in FreeEagle’s Table 4. We have included baseline results from K-arm's [9] Table 1, which is a baseline primarily focusing on this benchmark.
> We will ensure these details and the additional comparative analysis are explicitly included in the revised version to enhance clarity and comprehensibility.
>
>
> [1] Gu et al. Badnets 2017
>
> [2] Nguyen et al. Wanet 2021
>
> [3] Chen et al. Targeted 2017
>
> [4] Nguyen et al. Input-aware 2020
>
> [5] Barni et al. New backdoor 2019
>
> [6] Wang et al. Bppattack 2022
>
> [7] Li et al. Invisible 2021
>
> [8] Turner et al. Label-consistent 2019
>
> [9] Shen et al. Backdoor scanning 2021

---

> > ### Comment · Reviewer_3HiF · 2024-08-12
> >
> > Thank you for the clarification and additional experiments. The author should consider adding this content to the article. I have decided to raise my score.

---

> ### Author Response · Authors · 2024-08-12
> **Thank You for Your Positive Feedback**
>
> Thank you for your positive feedback and for considering a higher score for our work! We will ensure that these experiments are incorporated to further enhance the manuscript.
>
> Sincerely,
> The Authors

---

### Author Rebuttal · Authors · 2024-08-06

One of the common concerns raised by the reviewers was TRODO's performance against adaptive attacks. To address this concern, we conducted additional experiments and present the results here. Also there were concerns on the details of hard transformations, which we have also addressed in the following. We assure the reviewers that we will discuss these in our final manuscript and include the provided experiments.

>**TRODO under Adaptive Attack:**

We define two types of adaptive attacks targeting TRODO, which we will describe below:

TRODO operates on the implicit assumption that a classifier is more confident on in-distribution (ID) samples than on out-of-distribution (OOD) samples, as shown in the OOD detection literature. The maximum softmax probability (MSP) assigned by the classifier to an input sample is considered as its confidence level, referred to as the ID-Score in our paper.

An adaptive attacker knowing our defense mechanism and aiming to undermine the core principle of TRODO, we believe training a classifier with a new loss function that equalizes the ID-Score for both ID and OOD samples would be an adversarial approach to diminish our defense. As a result, we propose the following loss function for training a $k$-class classifier $f$:

$L_{adaptive1}$=$E_{(x, y) ∼ D_{\text{in}}}$  [ $- \log f_{y}(x) $]  $- λ_1 E_{(x, y) ∼ D_{\text{out}}}$  [ $H(U; f(x))  $] $+ λ_2 E_{(x, y) ∼ D_{\text{in}}}$  [ $H(U; f(x))  $]

Here, $x, y$ are data samples and their labels, $f_{y}(x)$ denotes $y$-th output of the classifier, $U$ is the uniform distribution over $k$ classes, and $H$ is the cross-entropy. The first term of $L_{adaptive1}$ is the classification term (cross-entropy), while the other terms force the classifier to decrease MSP (ID-Score) for ID samples while increasing it for OOD samples. We set $λ_1=λ_2 = 0.5$, inspired by [1], to balance the importance of the first term. By this loss function, we hope the ID-Score for both OOD and ID samples will change while the classifier's decisions for them remain fixed.

Moreover, we define another loss function for adaptive attacking. Our study relies on the difference in ID-Score between a benign OOD sample and a perturbed OOD sample as the scanning signature, where a higher signature indicates that the classifier is trojaned. In this adaptive attack, we explicitly target this by defining a new loss function that aims to decrease the signature for trojan classifiers, making it harder for our defense to distinguish them from clean ones:

$L_{adaptive2}$=$E_{(x, y) ∼ D_{\text{in}}}$  [ $- \log f_{y}(x) $]  $ -  λ_3 E_{(x, y) ∼ D_{\text{out}}}$  [ $H(f(x); f(x^*))  $]

Here, $x^*$ denotes the adversarially perturbed sample.

We recreated our benchmark evaluation set, including both clean and trojaned classifiers, and trained trojan models using $L_{adaptive1}$ and $L_{adaptive2} $ and examined TRODO's performance in distinguishing them.

We should note that in $L_{adaptive1}$ although the second and third terms act as regularizers to disrupt our core principle, the trojaned classifier still assigns relatively high confidence to ID samples as it is forced to classify IDs properly. Moreover, the designed loss functions $L_{adaptive1}$ and $L_{adaptive2}$ make the classifiers overfit to utilized OODs, as there is no pattern behind that loss function for to be learned by the model. Furthermore, we use a random process to create OODs in TRODO (selecting random hard transformations to obtain $G(.)$ ). This offers some remedy against such adaptive attacks.

Here are the results for these experiments:

|**LabelMapping**|**Loss**|**MNIST**|**CIFAR10**|**GTSRB**|**CIFAR100**|**Pubfig**|***Avg.***|
|-|-|-|-|-|-|-|-|
|||ACC/ACC\*|ACC/ACC\*|ACC/ACC\*|ACC/ACC\*|ACC/ACC\*|**ACC/ACC\***|
|**All-to-One**|||||||
||$L_{default}$|91.2/89.6|91.0/88.4|96.6/93.2|86.7/82.5|88.1/83.0|**90.7/87.3**|
||$L_{adaptive1}$|87.1/84.8|87.1/84.5|91.7/89.2|79.8/78.5|81.0/79.8|**85.3/83.4**|
||$L_{adaptive2}$|87.3/86.3|88.1/86.6|93.0/90.8|83.3/81.0|83.7/81.1|**87.1/85.2**|
|**All-to-All**|||||||
||$L_{default}$|90.0/87.4|89.3/87.5|92.6/89.1|82.4/85.0|83.2/80.9|**87.5/86.0**|
||$L_{adaptive1}$|76.9/74.8|78.2/76.8|82.1/80.4|73.0/71.3|69.2/67.0|**75.9/74.1**|
||$L_{adaptive2}$|84.4/83.3|85.5/83.8|85.6/84.1|78.5/77.3|79.7/78.4|**82.7/81.4**|

----
>**Details of Hard Transformations:**

We define a set $\mathcal{T} = $ { $ \{T_i\}$ } , with each $T_i$ representing a specific type of hard augmentation. For each ID sample $x$, a random subset of $k$ members from $\mathcal{T}$ is selected and permuted, $\{T_{j_1}, T_{j_2}, \ldots, T_{j_k}\}$, and the transformations are sequentially applied, resulting in $T_{j_k}(\ldots T_{j_1}(x))$.

Each transformed training sample $x$ becomes a crafted OOD $x'$, with the transformation process denoted by $G(\cdot)$, i.e., $x' = G(x)$, where $G(\cdot) = T_{j_k}(\ldots T_{j_1}(x))$. We avoid using $k=1$, which would apply only a single hard transformation, because in some cases, the hard transformation does not significantly alter the semantics. For instance, applying rotation on a "Car" image yields an OOD as it is rare in natural images. However, some semantics are rotation invariant, such as "Flower" images. Therefore, we use $k>1$, and by rule of thumb, $k=3$ has been used. This ensures that the output of $G(\cdot)$ is sufficiently shifted from the ID.

For creating the set of hard transformations, we use techniques that have been demonstrated to be harmful in the literature, including Jigsaw, Random Erasing, CutPaste, Rotation, Extreme Blurring, Intense Random Cropping, Noise Injection, Extreme Cropping, Mixup, Cutout, CutMix, and Elastic Transform [2,3,4,5]. These are the methods investigated in the literature for crafting OODs. Fig5 provides some examples of crafted OODs.

[1] Hendrycks et al. Deep Anomaly 2019

[2] Miyai Rethinking 2023

[3] Kalantidis Hard 2020

[4] Li Cutpaste 2021

[5] Sinha Negative 2021

---

### Author Response · Authors · 2024-08-13

Dear Reviewer d7KJ,


As we approach the discussion period deadline, we kindly request that you review our rebuttal, in which we have aimed to thoroughly address your comments. If you have any further concerns, we would be most grateful if you could bring them to our attention, and we would be pleased to discuss them.

Sincerely,
The Authors

---

### Decision · Program_Chairs · 2024-09-25

**Decision:**

Accept (poster)

**Comment:**

The paper deals with the security of machine learning (ML) methods. Specifically, the paper proposes "TRODO", a novel defense to address the problem of "trojaned/backdoored" ML models, whose training process was compromised with adversarial manipulations that have been deliberatedly introduced by attackers. Despite being a well-known security issue, an effective solution to identifying "trojaned" models has yet to be found and TRODO seeks to overcome the limitations of prior work (such as, e.g., having poor performance when the ML model is subject to adversarial training on the poisoned dataset). TRODO is claimed to provide a "general" solution, allegedly being "agnostic" to the type of trojaning attack being employed.

The manuscript was reviewed by 6 referees with varying expertise in the application domain tackled by the paper. The reviewers praised the magnitude of the evaluation, the results, the underlying idea, the presentation/writing quality, and the fact that the paper also sheds light on the theoretical rationale behind the effectiveness of the proposed method. However, the reviewers also expressed some concerns, such as an unclear threat model, the lack of comparison with some state-of-the-art anti-trojaning methods, and the lack of consideration of an adaptive attacker.

Three reviewers, after interacting with the authors, recommend that the paper is worthy of a "6: Weak Accept", whereas another reviewer recommends a "5: Borderline Accept". In contrast, one reviewer provided a very critical review, recommending a "3: Reject" and despite actively participating in the discussion (both with the authors, as well as with other reviewers), their recommendation did not change. Finally, one reviewer recommends a "4: Borderline Reject", and is confident of the assessment, but did not acknowledge to having read the authors' response---which, in my opinion, have addressed the concerns expressed by this reviewer.

Due to the above, I am fairly confident that this paper should be accepted: even though the _submitted manuscript_ was providing a contribution that was not yet ready to be presented at a top-tier venue such as NeurIPS, the authors' efforts in responding to the reviewers' valid remarks have shed further light on the claimed contribution, allowing it to meet the expectations of NeurIPS.

===============

To be more precise, the original submission provided a contribution that was deemed to be theoretically sound, correctly evaluated, and properly presented/motivated. Doing all of this in a (relatively) short manuscript is challenging. However, some important elements were left out, such as the effectiveness of the proposed method against an adaptive adversary (as pointed out by various reviewers), which is a necessary evaluation to carry out whenever a new "defense" is proposed. The authors have acknowledged such a shortcoming (including admitting that some attackers may completely bypass the method), and have also proceeded to carry out an evaluation considering some attacks that can be staged against TRODO.

From a security standpoint, it is well-known that "no defense is foolproof". Hence, it should not come at a surprise that TRODO cannot work against very strong attackers. The author-reviewer interactions enabled to clarify this holds also for TRODO, thereby "warning" downstream users/researchers of the limitations of TRODO in the real world. For this reason, I think that there are no grounds to reject the paper. However, the authors are encouraged to enhance their manuscript by acknowledging these limitations (potentially by toning down the overall effectiveness of TRODO) and incorporating the new experiments and clarifications that emerged during the peer review (potentially by referring to the appendix---or even to the very discussion that transpired here on OpenReview!)